# Target Propagation via Regularized Inversion for Recurrent Neural Networks

**Vincent Roulet**[*]
*Department of Statistics*
*University of Washington*

*vroulet@uw.edu*

**Zaid Harchaoui**
*Department of Statistics*
*University of Washington*

*zaid@uw.edu*

**Reviewed on OpenReview:** *https: // openreview. net/ forum? id= Q5vdEJyhA8*

## Abstract

Target Propagation (TP) algorithms compute targets instead of gradients along neural networks and propagate them backward in a way that is similar to yet different than gradient back-propagation (BP). The idea initially appeared as a perturbative alternative to BP that may improve gradient evaluation accuracy when training multi-layer neural networks (LeCun, 1985) and has gained popularity as a biologically plausible counterpart of BP. However, there have been many variations of TP, and a simple version of TP still remains worthwhile. Revisiting the insights of LeCun (1985) and Lee et al. (2015), we present a simple version of TP based on regularized inversions of layers of recurrent neural networks. The proposed TP algorithm is easily implementable in a differentiable programming framework. We illustrate the algorithm with recurrent neural networks on long sequences in various sequence modeling problems and delineate the regimes in which the computational complexity of TP can be attractive compared to BP.

## 1 Introduction

Target Propagation (TP) algorithms can be seen as perturbative learning alternatives to the gradient back-propagation algorithm, where virtual targets are propagated backward instead of gradients (LeCun, 1985; Rohwer, 1989; Mirowski & LeCun, 2009; Bengio, 2014; Lee et al., 2015; Meulemans et al., 2020; Manchev & Spratling, 2020). A high-level summary is presented in Fig. 1: while gradient back-propagation considers storing intermediate gradients in a forward pass, TP algorithms proceed by computing and storing approximate inverses. The approximate inverses are then used to pass targets backward along the graph of computations to finally yield a weight update for stochastic learning.

TP aims to take advantage of the availability of approximate inverses to compute better descent directions for the objective at hand. Bengio et al. (2013) and Bengio (2020) argued that the approach could be relevant for problems involving multiple compositions such as the training of Recurrent Neural Networks (RNNs), which generally suffer from the phenomenon of exploding or vanishing gradients (Hochreiter, 1998; Bengio et al., 1994; Schmidhuber, 1992). This perspective was studied theoretically for Difference Target Propagation (DTP), a modern variant of TP, that was related to an approximate Gauss-Newton method, suggesting interesting venues to explain the benefits of TP (Bengio, 2020; Meulemans et al., 2020; 2021; Ernoult et al., 2022; Fairbank et al., 2022).

Another motivation for TP comes from neuroscience to propose biologically plausible learning processes in the brain and overcome the biological implausibility of gradient Back-Propagation (BP) (Ernoult et al.,

---

[*]Now at Google.

Fig. 1: Our implementation of target propagation uses the linearization of gradient inverses instead of gradients in a backward pass akin to gradient back-propagation.

2022). The biological plausibility of TP required its actual implementation to use several approximations such as learning inverses with reverse layers (Lee et al., 2015; Manchev & Spratling, 2020). However, it is unclear whether such reverse layers learn layer inverses during the training process, which hinders the study of the original idea of using approximate inverses.

To promote the original idea of TP and guide its implementation from both a biological and an optimization viewpoint, we develop here a simplified implementation of TP using analytical formulations. The proposed approach formalizes several heuristics used in previous implementations to highlight the key components that can make TP successful to, e.g., learning from long sequences. In our implementation of TP for recurrent neural networks, we consider approximating inverses via their variational formulation with analytic expressions. The analytic expression of our approximate inverse sheds light on the relevance of a regularization term which can be interpreted as inverting perturbed layers, an insight studied earlier by Lee et al. (2015) and Manchev & Spratling (2020) when learning reversed layers. In addition, by using analytical expressions, we can directly test recent interpretations of TP as an approximate Gauss-Newton method whose approximation comes from the learned reverse layers (Bengio, 2020; Meulemans et al., 2020; Ernoult et al., 2022).

In this spirit, we also use the interpretation of the DTP formula (Lee et al., 2015) as a finite difference approximation of a linearized regularized inverse to propose a smoother formula that can directly be integrated into a differentiable programming framework. We detail the computational complexity of the proposed implementation of TP and compare it to the one of BP, showing that the additional cost of computing inverses can be effectively amortized for very long sequences. Following the benchmark of Manchev & Spratling (2020), we observe that the proposed implementation of TP can perform better than classical gradient-based methods on several tasks involving RNNs. The code is made publicly available at `https://github.com/vroulet/tpri`.

**Related work.** The use of the gradient Back-Propagation (BP) algorithm to train deep networks was popularized by Rumelhart et al. (1986), though the algorithm had been used in several other fields before (Werbos, 1974; 1994; Griewank & Walther, 2008; Goodfellow et al., 2016). At the same time, alternatives to BP were proposed by, e.g., (LeCun, 1985, Section 2.2), by using "ideal states" propagated through the transpose of the weights and initialized at the desired outputs to define local learning objectives at each layer of the network. This idea was further explored in two different ways: either by redefining the training problem through the optimization of ideal states called "targets" as initiated by Rohwer (1989) or by redefining the backward dynamics of BP to back-propagate the desired outputs through approximate inverses as developed by, e.g., Bengio (2014) and Lee et al. (2015).

*Optimization in target space.* The training of deep networks can be seen as an optimization problem over both the weights of the layers and the intermediate states computed by the network, the two different sets of variables being constrained by the dynamics of the network. For a given set of weights, the intermediate states are entirely determined by the input of the network; that is the viewpoint adopted by BP by using forward-backward passes on the network (Rumelhart et al., 1986). Rohwer (1989) proposed to consider the intermediate states as the main variables of the network, which he called "targets", by computing for a fixed set of targets the optimal weights such that the dynamics are approximately satisfied while the input and

output of the network are fixed at the values given by the training set. The overall training objective depends only on targets that can be optimized by gradient descent on the error of reconstruction of the dynamics.

Such an approach has been re-proposed under different names by Atiya & Parlos (2000) and Castillo et al. (2006) and later cast as penalized formulation of the dynamical constraints by Carreira-Perpinan & Wang (2014) who exploited the resulting unconstrained formulation to decouple the training of each layer by alternate optimization. Taylor et al. (2016) and Gotmare et al. (2018) considered tackling the dynamical constraints by using a Lagrangian formulation of the problem and using an Alternative Direction Method of Multipliers (ADMM). In this spirit, Wiseman et al. (2017) used an ADMM-like algorithm for language modeling and reported disappointing experimental results.

On the other hand, Fairbank et al. (2022) revisited the original approach of Rohwer (1989) by considering a reconstruction error of the dynamics only based on its linear part and reported successful experimental results on several architectures. The main difficulty of the optimization in target space is that the total number of variables is a priori equal to the number of samples times the width and depth of the network. This difficulty was overcome by Fairbank et al. (2022) by considering optimizing targets defined on a subset of the training set, then propagating the information given by new samples by the chain rule. Taylor et al. (2016) overcame this issue by simplifying the optimization of the targets as one coordinate descent pass on the Lagrangian formulation which then resembles a back-propagation algorithm. In our work, we consider back-propagating targets to update weights rather than considering directly the targets as variables.

*Back-propagating targets or gradient surrogates.* The original idea of LeCun (1985) to use back-propagation mechanisms that could back-propagate ideal states through the network was revisited a few decades later by Bengio (2014) and Lee et al. (2015). Although directly using approximate inverses of the layers to back-propagate targets appeared unsuccessful, a slight variant called Difference Target Propagation (DTP), which stabilizes the inverses by means of a finite difference scheme explained in Sec. 2.2, matched roughly the performance of BP. We formalize this variant by using directly the Jacobian of the approximate inverse to test the relevance of the approach from an optimization viewpoint. DTP has been recently interpreted theoretically as an approximate Gauss-Newton method (Bengio, 2020; Meulemans et al., 2020; 2021), a viewpoint that we discuss in detail in Appendix C. On a practical side, DTP was successfully implemented for recurrent neural networks by Manchev & Spratling (2020). We follow the experimental benchmark done by Manchev & Spratling (2020) while formalizing some of the mechanisms of DTP in our work. Recently, Ahmad et al. (2020) and Dalm et al. (2021) considered using analytical inverses to implement TP and blend it with what they called a gradient-adjusted incremental formula. Yet, an additional orthogonality penalty is critical for their approach to work, while our approach can dispense of such additional penalty.

The back-propagation of gradients has also been bypassed by approximating the sensitivity of the output of the network to small perturbations by Le Cun et al. (1988) in order to update weights according to these sensitivities. Alternatively, gradient surrogates were approximated with "synthetic gradients" by Jaderberg et al. (2017) and Czarnecki et al. (2017) to decouple the backward pass of feed-forward deep networks and speed up the training process. Roulet & Harchaoui (2022) considered computing gradients of the Moreau envelope through the structure of the network and cast several previous works as instances of approximate computations of the gradients of the Moreau envelope. Rather than modifying the backward operations in the layers, one can also modify the weight updates for deep forward networks by using a regularized inverse as successfully implemented by Frerix et al. (2018). Recently, Amid et al. (2022) generalized the approach of Frerix et al. (2018) by considering alternative reconstruction losses, that is, alternative objectives for the weights to minimize for given targets by using matching losses on pre-activations or post-activations targets defined by a gradient step on these quantities.

*Biological plausibility.* A recurrent motivation for the development of TP algorithms has been the biological implausibility of BP (Crick, 1989). We summarize here some of the arguments against the biological plausibility of BP and refer the interested reader to, e.g., Bengio (2020) and Manchev & Spratling (2020) for a detailed discussion. First BP requires each neuron to emit two distinct signals (forward and backward) which has not been observed neurophysiologically (Roelfsema & Ooyen, 2005). Another issue is that BP solves the credit assignment problem (Minsky, 1961; Hinton et al., 1984) by using weight transport, implying a symmetric backward connectivity pattern, which is thought impossible in the brain (Lillicrap et al.,

2016). On the other hand, DTP may circumvent some of these issues such as weight transport by using different dynamics forward and backward. The credit assignment problem and other biological constraints to model learning schemes motivated several variations of TP or other learning schemes such as direct feedback alignment (Nøkland, 2016; Bartunov et al., 2018; Crafton et al., 2019; Lansdell et al., 2019; Akrout et al., 2019; Kunin et al., 2020; Ernoult et al., 2022). As mentioned earlier, our objective here is to complement the biological viewpoint on TP by taking an optimization viewpoint to understand its relevance in training performance and use these findings to serve as guidelines for biologically plausible implementations of TP.

**Notation.** For $f : \mathbb{R}^p \times \mathbb{R}^q \to \mathbb{R}^d$, the partial derivative of $f$ w.r.t. $x$ on $(x, y) \in \mathbb{R}^p \times \mathbb{R}^q$ is denoted $\partial_x f(x, y) = \left( \partial f^j(x, y) / \partial x_i \right)_{i,j} \in \mathbb{R}^{p \times d}$.

## 2 Target Propagation with Linearized Regularized Inverses

While TP was initially developed for multi-layer neural networks, we focus on its implementation for Recurrent Neural Networks (RNNs), as we shall follow the benchmark of Manchev & Spratling (2020) in the experiments. RNNs are also a canonical family of neural networks in which interesting phenomena arise in back-propagation algorithms.

**Problem setting.** A simple RNN parameterized by $\theta = (W_{hh}, W_{xh}, b_h, W_{hy}, b_y)$ maps a sequence of inputs $x_{1:\tau} = (x_1, \ldots, x_\tau)$ to an output $\hat{y} = F_\theta(x_{1:\tau})$ by computing hidden states $h_t \in \mathbb{R}^{d_h}$ corresponding to the inputs $x_t \in \mathbb{R}^{d_x}$. Formally, the output $\hat{y}$ and the hidden states $h_t$ are computed as an output operation following transition operations

$$\hat{y} = g_\theta(h_\tau) := s(W_{hy} h_\tau + b_y),$$
$$h_t = f_{\theta,t}(h_{t-1}) := a(W_{xh} x_t + W_{hh} h_{t-1} + b_h) \quad \text{for } t \in \{1, \ldots, \tau\},$$

where $s$ is, e.g., the soft-max function for classification tasks, $a$ is a non-linear operation such as the hyperbolic tangent function, and the initial hidden state is generally fixed as $h_0 = 0$. Given samples of sequence-output pairs $(x_{1:\tau}, y)$, the RNN is trained to minimize the error $\ell(y, F_\theta(x_{1:\tau}))$ of predicting $\hat{y} = F_\theta(x_{1:\tau})$ instead of $y$.

As one considers longer sequences, RNNs face the challenge of exploding/vanishing gradients $\partial F_\theta(x_{1:\tau}) / \partial h_t$ (Bengio & Frasconi, 1995); see Appendix A for more discussion. We acknowledge that specific parameterization-based strategies have been proposed to address this issue of exploding/vanishing gradients, such as orthonormal parameterizations of the weights (Arjovsky et al., 2016; Helfrich et al., 2018; Lezcano-Casado & Martınez-Rubio, 2019). The focus here is to simplify and understand TP as a back-propagation-type algorithm using RNNs as a workbench. Indeed, training RNNs is an optimization problem involving multiple compositions for which approximate inverses can easily be available. The framework could also be applied to, e.g., time-series or control models (Roulet et al., 2019).

Given the parameters $W_{hh}, W_{xh}, b_h$ of the transition operations, we can get approximate inverses of $f_{\theta,t}(h_{t-1})$ for all $t \in \{1, \ldots, \tau\}$, that yield optimization surrogates that can be better performing than the ones corresponding to regular gradients. We present below a *simple version* of TP based on *regularized analytic inverses* and *inverse linearizations*.

**Back-propagating targets.** The idea of TP is to compute virtual targets $v_t$ for each layer $t = \tau, \ldots, 1$ such that if the layers were able to match their corresponding target at time $t$, i.e., $f_{\theta,t}(h_{t-1}) \approx v_t$, the objective would decrease. The final target $v_\tau$ is computed as a gradient step on the loss w.r.t. $h_\tau$. The targets are back-propagated using an approximate inverse $f_{\theta,t}^\dagger$ of $f_{\theta,t}$ at each time step. detailed in Sec. 2.1.

Formally, consider an RNN that computed $\tau$ states $h_1, \ldots, h_\tau$ from a sequence $x_1, \ldots, x_\tau$ with associated output $y$. For a given stepsize $\gamma_h > 0$, we propose to back-propagate targets by computing

$$v_\tau = h_\tau - \gamma_h \partial_h \ell(y, g_\theta(h_\tau)), \tag{1}$$

$$v_{t-1} = h_{t-1} + \partial_h f_{\theta,t}^\dagger(h_t)^\top (v_t - h_t), \quad \text{for } t \in \{\tau, \ldots, 1\}. \tag{2}$$

The update rule (2) blends two ideas: i) regularized analytic inversions to approximate the inverse $f_{\theta,t}^{-1}$ by $f_{\theta,t}^{\dagger}$; ii) linear approximation, i.e., using the Jacobian of the approximate inverse instead of the difference target propagation formula proposed by Lee et al. (2015, Eq. 15). We shall also show that the update (2) puts in practice an insight from Bengio (2020) suggesting to use the inverse of the gradients in the spirit of a Gauss-Newton method. Once all targets are computed, the parameters of the transition operations are updated such that the outputs of $f_{\theta,t}$ at each time step move closer to the given target.

Formally, the update consists of a gradient step with stepsize $\gamma_\theta$ on the squared error between the targets and the current outputs, i.e., for $\theta_h \in \{W_{hh}, W_{xh}, b_h\}$,

$$\theta_h^{\text{next}} = \theta_h - \gamma_\theta \sum_{t=1}^{\tau} \partial_{\theta_h} \|f_{\theta,t}(h_{t-1}) - v_t\|_2^2/2. \tag{3}$$

As for the parameters $\theta_y = (W_{hy}, b_y)$, they are updated by a simple gradient step on the loss with the stepsize $\gamma_\theta$.

## 2.1 Regularized Analytic Inversion

To explore further the ideas of LeCun (1985) and Lee et al. (2015), we consider the variational definition of the inverse,

$$f_{\theta,t}^{-1}(v_t) = \operatorname*{argmin}_{v_{t-1} \in \mathbb{R}^{d_h}} \|f_{\theta,t}(v_{t-1}) - v_t\|_2^2 = \operatorname*{argmin}_{v_{t-1} \in \mathbb{R}^{d_h}} \|a(W_{xh}x_t + W_{hh}v_{t-1} + b_h) - v_t\|_2^2. \tag{4}$$

As long as $v_t$ belongs to the image $f_{\theta,t}(\mathbb{R}^{d_h})$ of $f_{\theta,t}$, this definition recovers exactly the inverse of $v_t$ by $f_{\theta,t}$. More generally, if $v_t \notin f_{\theta,t}(\mathbb{R}^{d_h})$, Eq. (4) computes the *best approximation* of the inverse in the sense of the Euclidean projection. For an injective activation function $a$ and $\theta_h = (W_{hh}, W_{xh}, b_h)$, the solution of (4) can easily be computed. Formally, for the sigmoid, the hyperbolic tangent, or the ReLU, their inverse can be obtained analytically for any $v_t \in a(\mathbb{R}^{d_h})$. So for $v_t \in a(\mathbb{R}^{d_h})$ and $W_{hh}$ full rank, we get

$$f_{\theta,t}^{-1}(v_t) = (W_{hh}^\top W_{hh})^{-1} W_{hh}^\top (a^{-1}(v_t) - W_{xh}x_t - b_h).$$

If $v_t \notin a(\mathbb{R}^{d_h})$, the minimizer of (4) is obtained by first projecting $v_t$ onto $a(\mathbb{R}^{d_h})$, before inverting the linear operation. To account for non-invertible matrices $W_{hh}$, we also add a regularization in the computation of the inverse. Overall we consider approximating the inverse of the layer by a regularized inverse of the form

$$f_{\theta,t}^{\dagger}(v_t) = (W_{hh}^\top W_{hh} + r\,\mathrm{I})^{-1} W_{hh}^\top (a^{-1}(\pi(v_t)) - W_{xh}x_t - b_h),$$

with a regularization parameter $r > 0$ and a projection $\pi$ onto $a(\mathbb{R}^{d_h})$ detailed in Appendix B.

**Analytic inversion vs. parameterized inversion.** Bengio (2014) and Manchev & Spratling (2020) parameterize the inverse as a reverse layer such that

$$f_{\theta,t}^{\dagger}(v_t) = \psi_{\theta',t}(v_t) := a(W_{xh}x_t + Vv_t + c),$$

and learn the parameters $\theta' = (V, c)$ for this reverse layer to approximate the inverse of the forward computations. Learning good approximations comes with a computational cost that can be better controlled by using analytic inversions presented above. The approach based on parameterized inverses may lack theoretical grounding, as pointed out by Bengio (2020), as we do not know how close the learned inverse is to the actual inverse throughout the training process. In contrast, the analytic inversion (4) is less *ad hoc*, it enables us to test directly the main idea of TP and it leads to competitive performance on real datasets.

The analytic formulation of the inverse gives simple insights into an approach with parameterized inverses. Namely, the analytic formula suggests parameterizing the reverse layer such that (i) the reverse activation is defined as the inverse of the activation, (ii) the layer uses a non-linear operation followed by a linear one instead of a linear operation followed by a non-linear one as used by, e.g., Manchev & Spratling (2020).

**Regularized inversion vs noise injection.** A potential issue with learning parameterized inverses as done by Manchev & Spratling (2020) and Lee et al. (2015) is that the inverse depends a priori on the current set of parameters of the transition function, which, in turn, changes along the training process. Manchev & Spratling (2020) and Lee et al. (2015) considered then stabilizing the learning process of the parameterized inverse by letting it minimize perturbed versions of the layers, an idea reminiscent of the early work of Le Cun et al. (1988).

Formally, the parameterized inverses are learned by performing gradient steps over $\theta'$ on $\sum_{t=1}^{\tau} \|\psi_{\theta',t}(f_{\theta,t}(h_{t-1} + \varepsilon_{t-1})) - (h_{t-1} + \varepsilon_{t-1})\|_2^2$ for $\epsilon_{t-1} \sim \mathcal{N}(0, \sigma^2 \mathrm{I})$ and $h_0, \ldots, h_{\tau-1}$ given by a forward pass on the RNN for a given sample and fixed parameters $\theta$. Such a strategy led to performance gains in the overall training (Manchev & Spratling, 2020, Section 3.1.2, Figure 7).

In our formulation, the regularization can be seen as a counterpart of the noise injection. Namely, consider the inverse of averaged perturbed layers, i.e, the minimizer in $v_{t-1}$ of $\mathbb{E}_{z \sim \mathcal{N}(0, \sigma^2 \mathrm{I})} \|f_{\theta+z,t}(v_{t-1}) - v_t\|_2^2$. As shown in Appendix C, the regularized inverse $f_{\theta,t}^{\dagger}$ minimizes an upper-bound of this objective, namely, for $\sigma^2 = r/d_h$ and $v_t \in a(\mathbb{R}^{d_h})$,

$$f_{\theta,t}^{\dagger}(v_t) = \underset{v_{t-1} \in \mathbb{R}^{d_h}}{\mathrm{argmin}} \ \mathbb{E}_{z \sim \mathcal{N}(0, \sigma^2 \mathrm{I})} \|a^{-1}(f_{\theta+z,t}(v_{t-1})) - a^{-1}(v_t)\|_2^2,$$

where $\mathbb{E}_{z \sim \mathcal{N}(0, \sigma^2 \mathrm{I})} \|a^{-1}(f_{\theta+z,t}(v_{t-1})) - a^{-1}(v_t)\|_2^2 \geq \ell_a \mathbb{E}_{z \sim \mathcal{N}(0, \sigma^2 \mathrm{I})} \|f_{\theta+z,t}(v_{t-1}) - v_t\|_2^2$ for a $\ell_a$-Lipschitz-continuous activation $a$ and the approximation error of using the upper-bound is quantified in Appendix C.

The regularization $r$ can then be interpreted as a hyper-parameter that ensures that the inverses computed for the current parameters are also approximately valid for the updated parameters as long as $\|\theta_h^{\mathrm{next}} - \theta_h\|_2^2 \leq \sigma^2 = r/d_h$, which hints that the regularization $r$ may be chosen such that $r$ is larger than a constant times $\gamma_\theta$.

## 2.2 Linearized Inversion

Earlier instances of TP used directly approximate inverses of the network layers such that the TP update formula would read $v_{t-1} = f_{\theta,t}^{\dagger}(v_t)$ in (2). Yet, we are unaware of a successful implementation of TP using directly the inverses. To circumvent this issue, Lee et al. (2015) proposed the *Difference Target Propagation* (DTP) formula

$$v_{t-1} = h_{t-1} + f_{\theta,t}^{\dagger}(v_t) - f_{\theta,t}^{\dagger}(h_t).$$

If the inverses were exact, the DTP formula would reduce to $v_{t-1} = f_{\theta,t}^{-1}(v_t)$. Lee et al. (2015) introduced the DTP formula to mitigate the approximation error of the inverses by parameterized layers. The DTP formula was recently interpreted as an approximate Gauss-Newton method (Meulemans et al., 2020; Bengio, 2020) by observing that it can be approximated by the linearization of the inverse as

$$f_{\theta,t}^{\dagger}(v_t) - f_{\theta,t}^{\dagger}(h_t) = \partial_h f_{\theta,t}^{\dagger}(h_t)^{\top}(v_t - h_t) + O(\|v_t - h_t\|_2^2).$$

Our implementation puts in practice this insight. We show in Appendix D that the linearization we propose (2) leads to slightly better training curves than the finite-difference approximation. Moreover, our linearized formula enables a simple implementation of TP in a differentiable programming framework and simplifies its comparison with gradient back-propagation from a computational viewpoint.

# 3  Gradient Back-Propagation versus Target Propagation

**Graph of computations.** Gradient back-propagation and target propagation both compute an update direction for the objective at hand. The difference lies in the oracles computed and stored in the forward pass, while the graph of computations remains the same. To clarify this view, we reformulate TP in terms

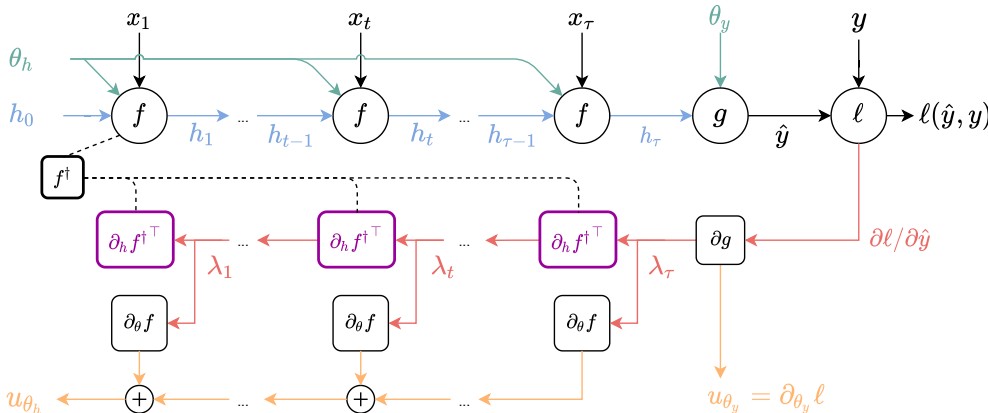

Fig. 2: The graph of computations of target propagation is the same as the one of gradient back-propagation except that $f^\dagger$ needs to be computed and Jacobian of the inverses, $\partial_h f^{\dagger \top}$ are used instead of gradients $\partial_h f$ in the transition operations.

of displacements $\lambda_t = v_t - h_t$ such that Eq. (1), (2) and (3) read

$$\lambda_\tau = -\gamma_h \partial_h \ell(y, g_\theta(h_\tau)), \qquad \lambda_{t-1} = \partial_h f_{\theta,t}^\dagger(h_t)^\top \lambda_t, \quad \text{for } t \in \{\tau, \dots, 1\},$$

$$u_{\theta_h} = \sum_{t=1}^{\tau} \partial_{\theta_h} f_{\theta,t}(h_{t-1}) \lambda_t, \qquad \theta_h^{\text{next}} = \theta_h + \gamma_h u_{\theta_h}.$$

TP amounts then to computing an update direction $u_{\theta_h}$ for the parameters $\theta_h$ with a graph of computations, illustrated in Fig. 2, analogous to that of BP illustrated in Appendix A. The difference lies in the use of the Jacobian of the inverse

$$\partial_h f_{\theta,t}^\dagger(h_t)^\top \quad \text{instead of} \quad \partial_h f_{\theta,t}(h_{t-1}).$$

The implementation of TP with the formula (2) can be done in a differentiable programming framework, where, rather than computing the gradient of the layer, one evaluates the inverse and keep the Jacobian of the inverse. With the precise graph of computation of TP and BP, we can compare their computational complexity explicitly and bound the difference in the directions they output.

**Arithmetic complexity.** Clearly, the space complexities of BP and our implementation of TP are the same since the Jacobians of the inverse and the original gradients have the same size. In terms of time complexity, TP appears at first glance to introduce an important overhead since it requires the computation of some inverses. However, a close inspection of the formula of the regularized inverse reveals that a matrix inversion needs to be computed only once for all time steps. Therefore the cost of the inversion may be amortized if the length of the sequence is particularly long.

Formally, the time complexity of the forward-backward pass of gradient back-propagation is essentially driven by matrix-vector products, i.e.,

$$\mathcal{T}_{\text{BP}} = \sum_{t=1}^{\tau} \Bigg[ \underbrace{\mathcal{T}(f_{\theta,t}) + \mathcal{T}(\partial_h f_{\theta,t}) + \mathcal{T}(\partial_{\theta_h} f_{\theta,t})}_{\text{Forward}} + \underbrace{\mathcal{T}(\partial_h f_{\theta,t}(h_{t-1})) + \mathcal{T}(\partial_{\theta_h} f_{\theta,t}(h_{t-1}))}_{\text{Backward}} \Bigg]$$

$$\approx \tau(d_x d_h + d_h{}^2) + \tau d_h{}^2,$$

where $d_x$ is the dimension of the inputs $x_t$, $d_h$ is the dimension of the hidden states $h_t$, for a function $f$ we denote by $\mathcal{T}(f)$ the time complexity to evaluate $f$ and we consider, e.g., $\partial_\theta f_{\theta,t}(h_{t-1})$ as the linear function $\lambda \to \partial_\theta f_{\theta,t}(h_{t-1})\lambda$.

On the other hand, the time complexity of target propagation is

$$\mathcal{T}_{\mathrm{TP}} = \sum_{t=1}^{\tau} \left[ \underbrace{\mathcal{T}(f_{\theta,t}) + \mathcal{T}(f_{\theta,t}^{\dagger}) + \mathcal{T}(\partial_{\theta_h} f_{\theta,t}) + \mathcal{T}(\partial_h f_{\theta,t}^{\dagger})}_{\text{Forward}} + \underbrace{\mathcal{T}(\partial_h f_{\theta,t}^{\dagger}(h_t)^{\top}) + \mathcal{T}(\partial_{\theta_h} f_{\theta,t}(h_{t-1}))}_{\text{Backward}} \right] + \mathcal{P}(f_{\theta,t}^{\dagger}),$$

where $\mathcal{P}(f_{\theta,t}^{\dagger})$ is the cost of encoding the inverse, which, in our case, amounts to the cost of encoding $c_{\theta} : z \to (W_{hh}^{\top} W_{hh} + r\,\mathrm{I})^{-1} W_{hh}^{\top} z$, such that our regularized inverse can be computed as $f_{\theta,t}^{\dagger}(v_t) = c_{\theta}(a^{-1}(v_t) - W_{xh} x_t + b_h)$. Encoding $c_{\theta}$ comes at the cost of inverting one matrix of size $d_h$. Therefore, the time-complexity of TP can be estimated as

$$\mathcal{T}_{\mathrm{TP}} \approx d_h^3 + \tau(d_x d_h + d_h^2) + \tau d_h^2 \approx \mathcal{T}_{\mathrm{BP}} \quad \text{if } \tau \geq d_h.$$

So for long sequences whose length is larger than the dimension of the hidden states, the cost of TP with regularized inverses is approximately the same as the cost of BP. If a parameterized inverse was used rather than a regularized inverse, the cost of encoding the inverse would correspond to the cost of updating the reverse layers by, e.g., a stochastic gradient descent. This update has a cost similar to BP. However, it is unclear whether these updates get us close to the actual inverses.

**Bounding the difference between target propagation and gradient back-propagation.** As the computational graphs of BP and TP are the same, we can bound the difference between the oracles returned by both methods. First, note that the updates of the parameters of the output functions are the same since, in TP, gradient steps of the loss are used to update these parameters. The difference between TP and BP lies in the updates with respect to the parameters of the transition operations.

For BP, the update direction is computed by chain rule as

$$\partial_{\theta_h} \ell\,(y, F_{\theta}(x_{1:\tau})) = \sum_{t=1}^{\tau} \partial_{\theta_h} f_{\theta,t}(h_{t-1}) \frac{\partial h_{\tau}}{\partial h_t} \partial_{h_{\tau}} \ell(y, g_{\theta}(h_{\tau})),$$

where the term $\partial h_{\tau}/\partial h_t$ decomposes along the time steps as $\partial h_{\tau}/\partial h_t = \prod_{s=t+1}^{\tau} \partial_{h_{s-1}} f_{\theta,s}(h_{s-1})$. The direction computed by TP has the same structure, namely it can be decomposed for $\gamma_h = 1$ as

$$u_{\theta_h} = \sum_{t=1}^{\tau} \partial_{\theta_h} f_{\theta,t}(h_{t-1}) \frac{\hat{\partial} h_{\tau}}{\hat{\partial} h_t} \partial_{h_{\tau}} \ell(y, g_{\theta}(h_{\tau})),$$

where $\hat{\partial} h_{\tau}/\hat{\partial} h_t = \prod_{s=t+1}^{\tau} \partial_{h_s} f_{\theta,s}^{\dagger}(h_s)^{\top}$. We can then bound the difference between the directions given by BP or TP as formally stated in the following lemma.

**Lemma 3.1.** *The difference between the oracle returned by gradient back-propagation $\partial_{\theta_h} \ell\,(y, F_{\theta}(x_{1:\tau}))$ and the oracle returned by target propagation $u_{\theta_h}$ can be bounded as*

$$\|\partial_{\theta_h} \ell\,(y, F_{\theta}(x_{1:\tau})) - u_{\theta_h}\| \leq \sum_{t=1}^{\tau} c_t \|\partial_{\theta_h} f_{\theta,t}(h_{t-1})\| \|\partial_{h_{\tau}} \ell(y, g_{\theta}(h_{\tau}))\| \sup_{t=1,\ldots,\tau} \|\partial_{h_{t-1}} f_{\theta,t}(h_{t-1}) - \partial_{h_t} f_{\theta,t}^{\dagger}(h_t)^{\top}\|,$$

*where $c_t = \sum_{s=0}^{t-1} a^s b^{t-1-s}$ with $a = \sup_{t=1,\ldots\tau} \|\partial_{h_{t-1}} f_{\theta,t}(h_{t-1})\|, b = \sup_{t=1,\ldots\tau} \|\partial_{h_t} f_{\theta,t}^{\dagger}(h_t)^{\top}\|$.*

*For regularized inverses, we have, denoting $z_t = W_{xh} x_t + W_{hh} h_{t-1} + b_h$,*

$$\|\partial_{h_{t-1}} f_{\theta,t}(h_{t-1}) - \partial_{h_t} f_{\theta,t}^{\dagger}(h_t)^{\top}\| \leq \|W_{hh}^{\top}\| \left( \|\nabla a(z_t) - \nabla a(z_t)^{-1}\| + \|\mathrm{I} - (W_{hh}^{\top} W_{hh} + r\,\mathrm{I})^{-1}\| \|\nabla a(z_t)^{-1}\| \right).$$

For the two oracles to be close, we need the preactivation $z_t = W_{xh} x_t + W_{hh} h_{t-1} + b_h$ to lie in the region of the activation function that is close to being linear s.t. $\nabla a(z_t) \approx \mathrm{I}$. We also need $(W_{hh}^{\top} W_{hh} + r\,\mathrm{I})^{-1}$ to be close to the identity which can be the case if, e.g., $r = 0$ and the weight matrices $W_{hh}$ were orthonormal. By initializing the weight matrices as orthonormal matrices, the differences between the two oracles can be close in the first iterations. However, in the long term, target propagation appears to give potentially better oracles, as shown in the experiments below. From a theoretical standpoint, the above lemma can be used to obtain convergence guarantees to stationary points of TP up to the approximation error of BP by TP as shown in Appendix C.

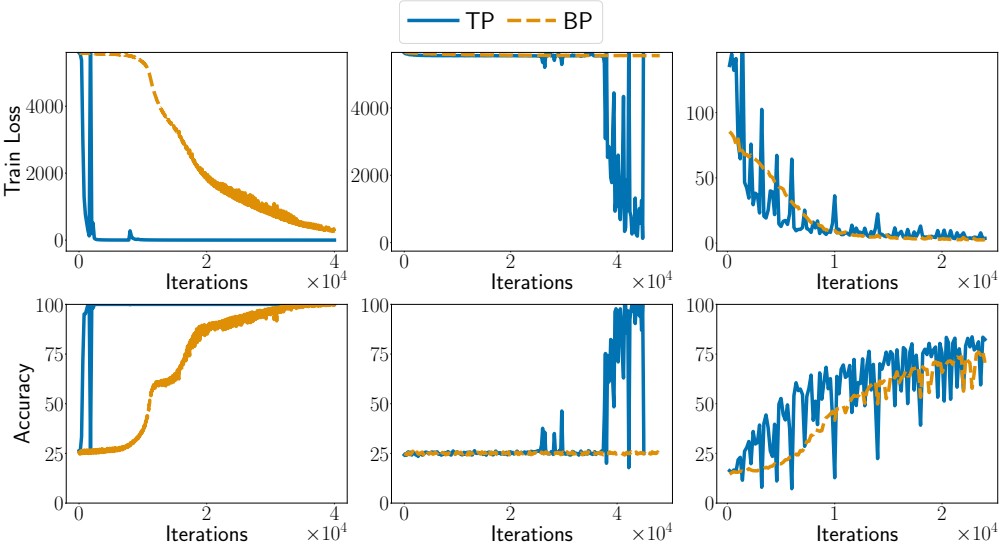

Fig. 3: Temporal order problem $T = 60$, Temporal order problem $T = 120$, Adding problem $T = 30$.

**Testing the interpretation of TP as a Gauss-Newton method.** Recently target propagation has been interpreted as an approximate Gauss-Newton (GN) method, by considering that the DTP formula approximates the linearization of the inverse, which itself is a priori equal to the inverse of the gradients (Bengio, 2020; Meulemans et al., 2020; 2021). Namely, provided that $f_{\theta,t}^\dagger(f_{\theta,t}(h_{t-1})) \approx h_{t-1}$ such that $\partial_{h_{t-1}} f_{\theta,t}(h_{t-1}) \partial_{h_t} f_{\theta,t}^\dagger(h_t) \approx \mathrm{I}$, we have

$$\partial_{h_t} f_{\theta,t}^\dagger(h_t) \approx \left(\partial_{h_{t-1}} f_{\theta,t}(h_{t-1})\right)^{-1}.$$

By composing the inverses of the gradients, we get an update similar to the one of GN. Namely, recall that if $n$ invertible functions $f_1, \ldots, f_n$ were composed to solve a least square problem of the form $\|f_n \circ \ldots \circ f_1(x) - y\|_2^2$, a GN update would take the form $x^{(k+1)} = x^{(k)} - \partial_{x_0} f_1(x_0)^{-\top} \ldots \partial_{x_{n-1}} f(x_{n-1})^{-\top}(x_n - y)$, where $x_t$ is defined iteratively as $x_0 = x^{(k)}$, $x_{t+1} = f_t(x_t)$. In other words, GN and TP share the idea of composing the inverse of gradients. However, numerous differences remain as detailed in Appendix C such as the fact that the gradient of the layer with respect to its parameters is not inverted in the usual implementation of TP and that the approximation error of GN by TP may grow exponentially with the length of the network. Note that even if TP was approximating GN, it is unclear whether GN updates are adapted to stochastic problems.

In any case, by using an analytical formula for the inverse, we can test this interpretation by using non-regularized inverses, which would amount to directly using the inverses of the gradients as in a GN method. If the success of TP could be explained by its interpretation as a GN method, we should observe efficient training curves when no regularization is added.

## 4 Experiments

In the following, we compare our simple TP approach, which we shall refer to as **TP**, to gradient Back-Propagation referred to as **BP**. We follow the experimental benchmark of Manchev & Spratling (2020) to which we add results for RNNs on CIFAR and for GRU networks on FashionMNIST. Additional experimental, details on the initialization and the hyper-parameter selection can be found in Appendix D.

**Data.** We consider two synthetic datasets generated to present training difficulties for RNNs and several real datasets consisting of scanning images pixel by pixel to classify them (Hochreiter & Schmidhuber, 1997; Le et al., 2015; Manchev & Spratling, 2020).

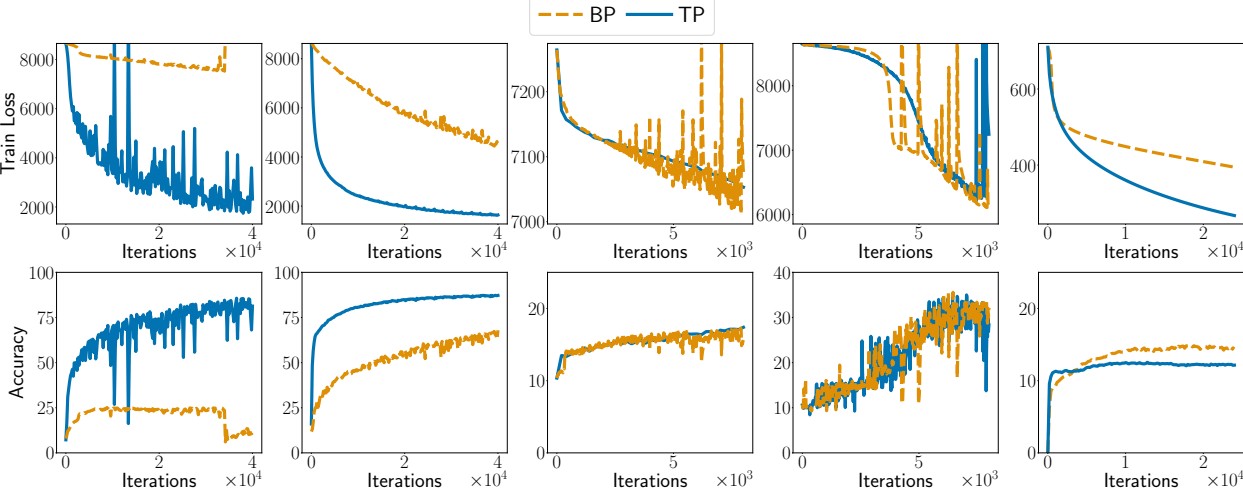

Fig. 4: Image classification pixel by pixel and word prediction. From left to right: MNIST, MNIST with permuted images, CIFAR10, FashionMNIST with GRU, Penn Treebank with RNN.

*Temporal order problem.* A sequence of length $T$ is generated using a set of randomly chosen symbols $\{a, b, c, d\}$. Two additional symbols $X$ and $Y$ are added at positions $t_1 \in [T/10, 2T/10]$ and $t_2 \in [4T/10, 5T/10]$. The network must predict the correct order of appearance of $X$ and $Y$ out of four possible choices $\{XX, XY, YX, YY\}$.

*Adding problem.* The input consists of two sequences: one is made of randomly chosen numbers from $[0, 1]$, and the other one is a binary sequence full of zeros except at positions $t_1 \in [1, T/10]$ and $t_2 \in [T/10, T/2]$. The second position acts as a marker for the time steps $t_1$ and $t_2$. The goal of the network is to output the mean of the two random numbers of the first sequence $(X_{t_1} + X_{t_2})/2$.

*Image classification pixel by pixel.* The inputs are images of (i) grayscale handwritten digits given in the database MNIST (LeCun & Cortes, 1998), (ii) colored objects from the database CIFAR10 (Krizhevsky, 2009) or (iii) grayscale images of clothes from the database FashionMNIST (Xiao et al., 2017). The images are scanned pixel by pixel and channel by channel for CIFAR10, and fed to a sequential network such as a simple RNN or a GRU network (Cho et al., 2014). The inputs are then sequences of $28 \times 28 = 784$ pixels for MNIST or FashionMNIST and $32 \times 32 \times 3 = 3072$ pixels for CIFAR with a very long-range dependency problem. We also consider permuting the images of MNIST by a fixed permutation before feeding them into the network, which gives potentially longer dependencies in the data.

*Word prediction.* Given a sequence of $t$ words from 10000 sentences of the Penn Treebank dataset (Marcinkiewicz, 1994), the task is to predict the $t + 1^{\text{th}}$ word among a vocabulary of 10 000 words. We consider sequences of length 64 and a fixed embedding of size 1024 of the words.

**Model.** In both synthetic settings, we consider randomly generated mini-batches of size 20, a simple RNN with hidden states of dimension 100, and hyperbolic tangent activation. For the temporal order problem, the output function uses a soft-max function on top of a linear operation, and the loss is the cross-entropy. For the adding problem, the output function is linear, the loss is the mean-squared error, and a sample is considered to be accurately predicted if the mean squared error is less than 0.04 as done by (Manchev & Spratling, 2020).

For the classification of images with sequential networks, we consider mini-batches of size 16 and a cross-entropy loss. For MNIST and CIFAR, we consider a simple RNN with hidden states of dimension 100, hyperbolic tangent activation, and a softmax output. For FashionMNIST, we consider a GRU network and adapted our implementation of target propagation to that case while using hidden states of dimension 100 and a softmax output, see Appendix B for the detailed implementation of TP for GRU networks.

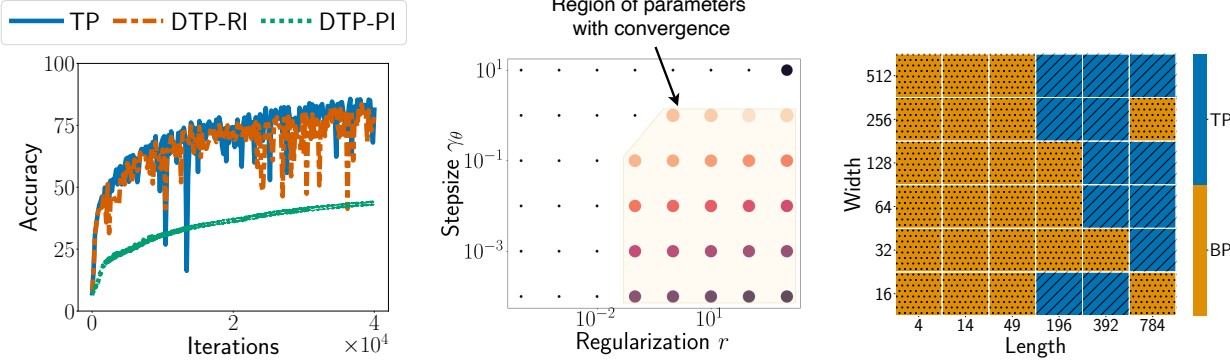

Fig. 5: Comp. of TP implem.    Fig. 6: Conv. w.r.t. stepsize & reg.    Fig. 7: Perf. vs width & length.

For the prediction of words, we consider an RNN that outputs a prediction at each time step. Our implementation of TP, in this case, consists in replacing the usual gradient back-propagation rule used in the computational scheme of gradient back-propagation by the Jacobian of the regularized inverse as explained in Sec. 3 and detailed in Appendix B. We use a simple RNN architecture with hyperbolic tangent activations for the transitions of hidden states of size 256, an output function consisting of a soft-max layer on top of a linear layer, and a cross-entropy loss.

**Performance comparisons.** In Fig. 3, we observe that TP performs better than BP on the temporal ordering problem: it is able to reach 100% accuracy in fewer iterations than BP for sequences of length 60 and, for sequences of length 120, it is still able to reach 100% accuracy in fewer than 40 000 iterations while BP is not. On the other hand, for the adding problem, TP performs less well than BP. The contrast in performance between the two synthetic tasks was also observed by (Manchev & Spratling, 2020) using difference target propagation with parameterized inverses. The main difference between these tasks is the nature of the outputs, which are binary for the temporal problem and continuous for the adding problem.

In Fig. 4, we observe that TP generally performs better or as well as BP for image classification tasks. For the MNIST dataset, it reaches around 74% accuracy after $4 \cdot 10^4$ iterations. This phenomenon is also observed with permuted images, where the optimization appears smoother, and TP obtains around 86% accuracy after $4 \cdot 10^4$ iterations and is still slightly faster than BP. On the CIFAR dataset, no algorithms appear to reach a significant accuracy, though TP is still faster. On the FashionMNIST dataset, where a GRU network is used, our implementation of TP performs on par with BP, which shows that our approach can be generalized to more complex networks than a simple RNN. For the word prediction task, our implementation of TP is still able to optimize well the objective in terms of training loss, outperforming BP in that regard, and performing slightly worse in terms of test accuracy.

**Comparison of different implementations of TP.** We evaluate the impact of using regularized inverses as opposed to parameterized inverses and linearized propagation as opposed to finite-difference-based propagation. The variant of target propagation with parameterized inverse and finite-difference propagation corresponds to the approach of Lee et al. (2015) recently implemented by Manchev & Spratling (2020) and referred to in Fig. 5 as DTP-PI. The variant of target propagation with regularized inverse and finite-difference propagation is referred to in Fig. 5 as DTP-RI. Recall that our approach involves regularized inverses and linearized propagation, referred to as TP. In Fig. 5, we observe that both TP and DTP-RI outperform DTP-PI, demonstrating the benefits of using regularized inverses. On the other hand, both TP and DTP-RI perform on par overall, with the former being slightly better for the given parameters. The linearized formula has the advantage to be easily adapted to other architectures such as a GRU network or an RNN with intermediate outputs as explained in Appendix B. Additional comparisons of our implementation of TP as opposed to the implementation of TP by Manchev & Spratling (2020) are presented in Appendix D.

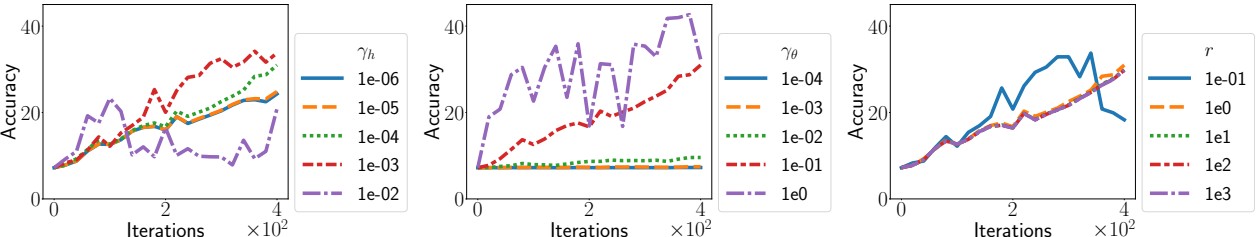

Fig. 8: Sensitivity analyses of the hyper-parameters of TP on MNIST pixel by pixel. From left to right: varying $\gamma_h$ for $\gamma_\theta = 10^{-1}$, $r=1$, varying $\gamma_\theta$ for $\gamma_h = 10^{-4}$, $r=1$, varying $r$ for $\gamma_h = 10^{-4}$, $\gamma_\theta = 10^{-1}$.

**Impact of the regularization term.** As mentioned in Sec. 3, by using an analytical formula to compute the inverse of the layers, we can question the interpretation of TP as a Gauss-Newton method, which would amount to TP without regularization. To understand the effect of the regularization term, we computed the area under the training loss curve of TP for 400 iterations on a $\log_{10}$ grid of varying step-sizes $\gamma_\theta$ and regularizations $r$ for a fixed stepsize $\gamma_h = 10^{-3}$. The results are presented in Fig. 6, where the smaller the area, the brighter the point and the absence of dots in the grid means that the algorithm diverged. Fig. 6 shows that without regularization we were not able to obtain convergence of the algorithm. Simply using the gradients of the inverse as in a Gauss-Newton method may not directly work for RNNs. Additional modifications of the method could be added to make target propagation closer to Gauss-Newton, such as inverting the layers with respect to their parameters as proposed by Bengio (2020). For now, the regularization appears to successfully handle the rationale of target propagation.

**Performance of TP and BP in terms of the length of the sequence.** In Fig. 7, we compare the performance of BP and TP in terms of accuracy after 400 iterations on the MNIST problem for various widths determined by the size of the hidden states and various lengths determined by the size of the inputs (i.e., we feed the RNN with $k$ pixels at a time, which gives a length $784/k$). Fig. 7 shows that TP is generally appropriate for long sequences, while BP remains more efficient for short sequences. TP may then be seen as an interesting alternative for dynamical problems which involve many discretization steps as in RNNs and related architectures.

**Sensitivity analyses.** In Fig. 8, we observe that the algorithm is mostly sensitive to the learning rate $\gamma_\theta$ while both the stepsize $\gamma_h$ used for the first target and the regularization $r$ admit a larger range of values ensuring fact increase of the accuracy. Note that for the regularization, while most values above $r = 1$ ensure a similar increase of accuracy, smaller values than $10^{-1}$ in the MNIST pixel-by-pixel experiment would not lead to convergent algorithms as explained previously in Fig. 6.

## Conclusion

We proposed a simple target propagation approach grounded in two important computational components, regularized inversion, and linearized propagation. The proposed approach also sheds light on previous insights and successful rules for target propagation. The code is made publicly available at `https://github.com/vroulet/tpri`. We have used target propagation within a stochastic gradient outer loop to train neural networks for a fair comparison to a stochastic gradient descent using gradient backpropagation. Developing adaptive stochastic gradient algorithms in the spirit of Adam that lead to boosts in performance when using target propagation instead of gradient backpropagation is an interesting avenue for future work. Continuous counterparts of target propagation in a neural ODE spirit is also an interesting avenue for future work.

**Acknowledgments.** This work was supported by NSF CCF-1740551, NSF DMS-1839371, the CIFAR program "Learning in Machines and Brains", and faculty research awards. We thank Nikolay Manchev for all the details he provided on his code. We also thank the reviewers and the action editor of TMLR for their insightful feedback that helped improve the content and the presentation of the paper.

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

## Appendix Outline

The Appendix is organized as follows.

## A   Gradient Back-Propagation in Recurrent Neural Networks

Given differentiable activation functions $a$, the training of recurrent neural networks is amenable to optimization by gradient descent. The gradients can be computed by gradient back-propagation implemented in modern differentiable programming software (Rumelhart et al., 1986; Werbos, 1994; Paszke et al., 2019; Abadi et al., 2015). The gradient back-propagation algorithm is illustrated in Fig. 9. Formally, the gradients are computed by the chain rule such that, for a sample $(y, x_{1:\tau})$, $\theta_h = (W_{hh}, W_{xh}, b_h)$, and $\hat{y} = F_\theta(x_{1:\tau})$ the predicted output,

$$\frac{\partial \ell\,(y, \hat{y})}{\partial \theta_h} = \sum_{t=1}^{\tau} \frac{\partial h_t}{\partial \theta_h} \frac{\partial h_\tau}{\partial h_t} \frac{\partial \hat{y}}{\partial h_\tau} \frac{\partial \ell}{\partial \hat{y}}.$$

The term $\partial h_\tau / \partial h_t$ decomposes along the time steps as

$$\frac{\partial h_\tau}{\partial h_t} = \prod_{s=t+1}^{\tau} \frac{\partial h_s}{\partial h_{s-1}}.$$

As $\tau$ grows, the norm of the term $\partial h_\tau / \partial h_t$ may then either increase to infinity (*exploding gradients*) or exponentially decrease to 0 (*vanishing gradients*). This phenomenon may prevent the RNN from learning from dependencies between temporally distant events (Hochreiter, 1998). Several solutions were proposed to tackle this issue, including changing the network architecture (Hochreiter & Schmidhuber, 1997), Hessian-free optimization (Sutskever et al., 2011), gradient clipping and regularization (Pascanu et al., 2012), or orthonormal parametrizations (Arjovsky et al., 2016; Helfrich et al., 2018; Lezcano-Casado & Martınez-Rubio, 2019). We consider here propagating targets instead of gradients as first presented by LeCun (1985) and recently revisited by Bengio (2014); Lee et al. (2015).

## B   Detailed Implementation

### B.1   Target Propagation for RNNs with final outputs

As detailed in Sec. 3, target propagation with linearized regularized inverses amounts to move along an update direction computed by a forward-backward algorithm akin to gradient propagation. The iterations of linearized target propagation are then summarized in Algo. 1 and Algo. 2.

In the implementation of the regularized inverses, since the inverses of activation functions such as the sigmoid or the tangent hyperbolic are numerically unstable, we consider projecting on a subset of $a(\mathbb{R}^{d_h})$. For the hyperbolic tangent, we clip the target to $[-1+\varepsilon, 1-\varepsilon]$ for $\varepsilon = 10^{-3}$. Concretely, for an hyperbolic tangent activation function, the projection is then $\pi(v) = (\min(\max(v_i, -1+\varepsilon), 1-\varepsilon))_{i=1}^{d_h}$ for $v \in \mathbb{R}^{d_h}$. To read Algo. 2, we recall our notations for $\theta = (W_{hh}, W_{xh}, b_h, W_{hy}, b_y)$:

$$g_\theta(h_\tau) = s(W_{hy}h_\tau + b_y), \tag{5}$$

$$f_{\theta,t}(h_{t-1}) = a(W_{xh}x_t + W_{hh}h_{t-1} + b_h), \tag{6}$$

$$f_{\theta,t}^\dagger(v_t) = (W_{hh}^\top W_{hh} + r\,\mathrm{I})^{-1} W_{hh}^\top (a^{-1}(\pi(v_t)) - W_{xh}x_t - b_h). \tag{7}$$

Note that Algo. 2 can also be used for mini-batches of sequence-output pairs since all operations are either element-wise or linear with respect to the sample of sequence-output pair.

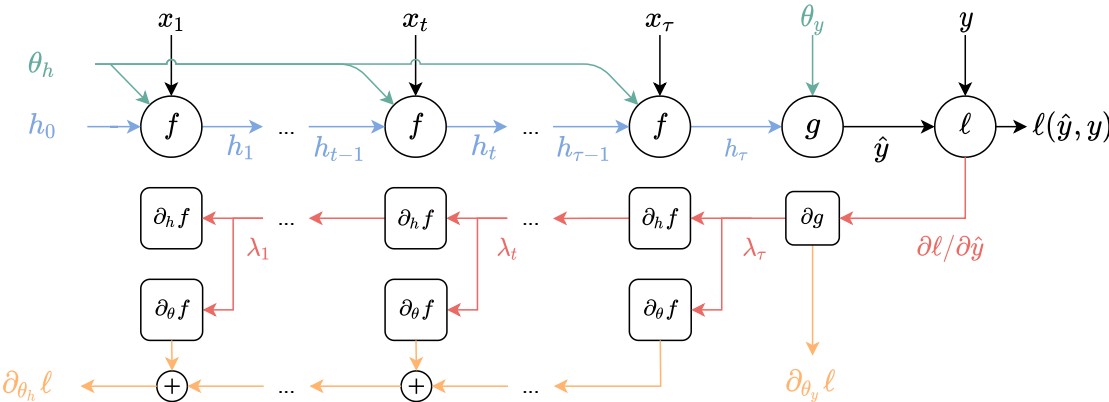

Fig. 9: Gradient back-propagation for RNNs.

---

**Algorithm 1** Stochastic learning with target propagation

---

1: **Inputs:** Initial parameters $\theta^{(0)} = (W_{hh}, W_{xh}, b_h, W_{hy}, b_y)$ of an RNN defined by Eq. (5) and (6), stepsize $\gamma_\theta$, total number of iterations $K$
2: **for** $k = 1 \ldots K$ **do**
3:     Draw a sample or a mini-batch of sequences-output pairs $(x_{1:\tau}, y)$.
4:     Compute
$$u_\theta = (u_{\theta_h}, u_{\theta_y}) = \text{TP}(\theta^{(k-1)}, x_{1:\tau}, y),$$
    where TP is Algo. 2
5:     Update the parameters as $\theta^{(k)} = \theta^{(k-1)} + \gamma_\theta u_\theta$.
6: **end for**

---

## B.2 Target Propagation for GRU Networks

### B.2.1 Formulation

Starting from $h_0 = 0$, given an input sequence $x_1, \ldots, x_\tau$, the GRU network (as implemented in Pytorch[1] (Paszke et al., 2019)), iterates for $t = 1, \ldots, \tau$,

$$m_t = f_{m,t}(h_{t-1}) := \sigma(W_{im}x_t + W_{hm}h_{t-1} + b_m) \tag{8}$$

$$z_t = f_{z,t}(h_{t-1}) := \sigma(W_{iz}x_t + W_{hz}h_{t-1} + b_z) \tag{9}$$

$$n_t = f_{n,t}(h_{t-1}, m_t) := \tanh(W_{in}x_t + b_{in} + m_t \odot (W_{hn}h_{t-1} + b_{hn})) \tag{10}$$

$$h_t = f_{h,t}(h_{t-1}, z_t, n_t) := (1 - z_t) \odot h_{t-1} + z_t \odot n_t, \tag{11}$$

where $\odot$ is the Hadamard product, $\sigma$ is the sigmoid function. In the following, we will denote simply $\theta = (\theta_m, \theta_z, \theta_n)$ the parameters of the network with

$$\theta_m = (W_{im}, W_{hm}, b_m), \qquad \theta_z = (W_{iz}, W_{hz}, b_z), \qquad \theta_n = (W_{in}, b_{in}, W_{hn}, b_{hn}).$$

The output of the network is, e.g., a soft-max operation on the hidden state computed at the last step (if applied to an image scanned pixel by pixel for example). See the main paper for the expression of the output in that case.

### B.2.2 Modifying the Chain Rule

The underlying idea of our implementation of target propagation in a differentiable programming framework is to mix classical gradients and Jacobians of the inverse of the functions. Denote for a given output loss $\mathcal{L}$

---

[1]Compared to https://pytorch.org/docs/stable/generated/torch.nn.GRU.html, we used a single variable $b_m = b_{im} + b_{hm}$, same for $b_z$.

---

**Algorithm 2** Proposed target propagation algorithm

---

1: **Parameters:** $\pi$ a projection onto a susbet of $a(\mathbb{R}^{d_h})$, stepsize $\gamma_h$, regularization $r$.
2: **Inputs:** Current parameters $\theta = (\theta_h, \theta_y)$ with $\theta_h = (W_{hh}, W_{xh}, b_h), \theta_y = (W_{hy}, b_y)$ of the RNN, sample of sequences-output pairs $(x_{1:\tau}, y)$.
3: Forward Pass:
4: Compute and store $V = (W_{hh}^\top W_{hh} + r\,\mathrm{I})^{-1} W_{hh}^\top$ giving access to $f_{\theta,t}^\dagger(v_t)$ defined in Eq. (7).
5: Initialize $h_0 = 0$.
6: **for** $t = 1, \ldots, \tau$ **do**
7:     Compute and store $h_t = f_{\theta,t}(h_{t-1}), \quad \partial_{\theta_h} f_{\theta,t}(h_{t-1}), \quad \partial_{h_t} f_{\theta,t}^\dagger(h_t)$.
8: **end for**
9: Compute and store $\ell(y, g_\theta(h_\tau)), \quad \partial_{\partial h_\tau} \ell(y, g_\theta(h_\tau)), \quad \partial_{\theta_y} \ell(y, g_\theta(h_\tau))$.
10: Backward Pass:
11: Define $\lambda_\tau = -\gamma_h \partial_{h_\tau} \ell(y, g_\theta(h_\tau)), \quad u_{\theta_y} = -\partial_{\theta_y} \ell(y, g_\theta(h_\tau))$.
12: **for** $t = \tau, \ldots, 1$ **do**
13:     Compute $\lambda_{t-1} = \partial_{h_t} f_{\theta,t}^\dagger(h_t)^\top \lambda_t$.
14: **end for**
15: **Outputs:** Update directions for $\theta_h, \theta_y$:

$$u_{\theta_h} = \sum_{t=1}^{\tau} \partial_{\theta_h} f_{\theta,t}(h_{t-1})\lambda_t, \qquad u_{\theta_y} = -\partial_{\theta_y} \ell(y, g_\theta(h_\tau)).$$

---

computed on a given mini-batch with the current parameters $\theta$, $\partial\hat{\mathcal{L}}/\partial\hat{h}_t$ the direction back-propagated by our implementation of target propagation until the step $h_t$. The directions for the parameters of the network can be output as

$$\frac{\hat{\partial}\mathcal{L}}{\hat{\partial}\theta} = \sum_{t=1}^{\tau} \frac{\partial h_t}{\partial \theta} \frac{\hat{\partial}\mathcal{L}}{\hat{\partial}h_t}.$$

The main task is to define $\hat{\partial}\mathcal{L}/\hat{\partial}h_{t-1}$ given $\hat{\partial}\mathcal{L}/\hat{\partial}h_t$ and appropriate regularized inverses. For that, we start with the chain rule for $\partial h_t/\partial h_{t-1}$ and we will replace some of the gradients by Jacobians of regularized inverses at some places.

**Classical chain rule.** We have

$$\frac{\partial h_t}{\partial h_{t-1}} = \left(-\frac{\partial z_t}{\partial h_{t-1}}\right) \mathrm{diag}(h_{t-1}) + \mathrm{I}\,\mathrm{diag}(1 - z_t) + \frac{\partial z_t}{\partial h_{t-1}} \mathrm{diag}(n_t) + \frac{\partial n_t}{\partial h_{t-1}} \mathrm{diag}(z_t) \tag{12}$$

$$= \mathrm{diag}(1 - z_t) + \frac{\partial z_t}{\partial h_{t-1}} \left(\mathrm{diag}(n_t) - \mathrm{diag}(h_{t-1})\right) + \frac{\partial n_t}{\partial h_{t-1}} \mathrm{diag}(z_t). \tag{13}$$

Now for $\partial n_t/\partial h_{t-1}$, we further decompose the function $f_{n,t}(h_{t-1})$ as $f_{n,t}(h_{t-1}) = g_t(m_t \odot a_t)$, with $g_t(u) = \tanh(W_{in}x_t + b_{in} + u)$ and $a_t = \ell(h_{t-1}) := W_{hn}h_{t-1} + b_{hn}$. We then have, denoting $p = m_t \odot a_t$

$$\frac{\partial n_t}{\partial h_{t-1}} = \left(\frac{\partial m_t}{\partial h_{t-1}} \mathrm{diag}(a_t) + \frac{\partial a_t}{\partial h_{t-1}} \mathrm{diag}(m_t)\right) \nabla g_t(p), \tag{14}$$

with $\nabla g_t(p) = \mathrm{diag}(\tanh'(W_{in}x_t + b_{in} + p))$.

**Inverses.** Now, the variables $z_t, m_t$ and $a_t$ are functions of $h_t$ that incorporate a linear operation and that can be inverted. Namely, we can define the following regularized inverses

$$f_{m,t}^\dagger(v_t) = (W_{hm}^\top W_{hm} + r\,\mathrm{I})^{-1} W_{hm}^\top(\sigma^{-1}(v_t) - W_{ir}x_t - b_m)$$

$$f_{z,t}^\dagger(v_t) = (W_{hz}^\top W_{hz} + r\,\mathrm{I})^{-1} W_{hz}^\top(\sigma^{-1}(v_t) - W_{iz}x_t - b_z)$$

$$\ell^\dagger(v_t) = (W_{hn}^\top W_{hn} + r\,\mathrm{I})^{-1} W_{hn}^\top(v_t - b_{hn}).$$

We can then do the following substitutions in Eq.(12) and (14),

$$\frac{\partial m_t}{\partial h_{t-1}} \leftarrow \frac{\hat{\partial} m_t}{\hat{\partial} h_{t-1}} = \partial f_{m,t}^\dagger(m_t)^\top, \quad \frac{\partial z_t}{\partial h_{t-1}} \leftarrow \frac{\hat{\partial} z_t}{\hat{\partial} h_{t-1}} = \partial f_{z,t}^\dagger(z_t)^\top, \quad \frac{\partial a_t}{\partial h_{t-1}} \leftarrow \frac{\hat{\partial} a_t}{\hat{\partial} h_{t-1}} = \partial \ell^\dagger(a_t)^\top,$$

to define the quantity back-propagated by target propagation.

Note that by taking the gradient of the inverse we can ignore the biases and the inputs. Namely, we have for example $\partial f_{m,t}^\dagger(m_t) = \text{diag}((\sigma^{-1})'(m_t)) W_{hm} (W_{hm}^\top W_{hm} + r\,\mathrm{I})^{-1}$, hence

$$\partial f_{m,t}^\dagger(m_t)^\top = (W_{hm}^\top W_{hm} + r\,\mathrm{I})^{-1} W_{hm}^\top \text{diag}((\sigma^{-1})'(m_t)).$$

The expression for $\partial f_{z,t}^\dagger(z_t)$ is identical. Since $\ell$ is affine, we have simply $\partial \ell^\dagger(a_t)^\top = (W_{hn}^\top W_{hn} + r\,\mathrm{I})^{-1} W_{hn}^\top$.

**Summary.** Combined together, we get, denoting $d_t = \frac{\hat{\partial} \mathcal{L}}{\partial h_t}$,

$$\begin{aligned}
\frac{\hat{\partial} \mathcal{L}}{\hat{\partial} h_{t-1}} &= (1 - z_t) \odot d_t + \partial f_{z,t}^\dagger(z_t)^\top ((n_t - h_{t-1}) \odot d_t) \\
&\quad + \partial f_{m,t}^\dagger(m_t)^\top (a_t \odot \tanh'(W_{in} x_t + b_{in} + u) \odot z_t \odot d_t) \\
&\quad + \partial \ell^\dagger(a_t)^\top (m_t \odot \tanh'(W_{in} x_t + b_{in} + u) \odot z_t \odot d_t) \\
&= (1 - z_t) \odot d_t + (W_{hz}^\top W_{hz} + r\,\mathrm{I})^{-1} W_{hz}^\top \left((\sigma^{-1})'(z_t) \odot (n_t - h_{t-1}) \odot d_t\right) \\
&\quad + (W_{hm}^\top W_{hm} + r\,\mathrm{I})^{-1} W_{hm}^\top \left((\sigma^{-1})'(m_t) \odot a_t \odot \tanh'(W_{in} x_t + b_{in} + u) \odot z_t \odot d_t\right) \\
&\quad + (W_{hn}^\top W_{hn} + r\,\mathrm{I})^{-1} W_{hn}^\top (m_t \odot \tanh'(W_{in} x_t + b_{in} + u) \odot z_t \odot d_t).
\end{aligned}$$

This provides a rule to propagate targets through linearized regularized inverses.

### B.3 Target Propagation for RNNs for Word Prediction

For the word prediction task, we consider a RNN that outputs a prediction at each time-step. Namely, the input-output samples consist in two sequences of words $(x_{1:\tau}, y_{1:\tau})$ where $y_t$ is the word following $x_t$ in a sequence of words. The RNN consists in first embedding the input words $x_t$ into a finite dimensional vector $\tilde{x}_t = D x_t$ with $D \in \mathbb{R}^{d_x \times v}$ where $v$ is the vocabulary size and $d_x$ is the embedding dimension given that $x$ is represented by a one-hot vector of size $v$. The RNN treats the embedded input as described in Sec. 2 and outputs a prediction at each time-step. Namely, the RNN can be summarized as outputting a sequence of predictions $\hat{y}_{1:\tau}$ from a sequence of inputs $x_{1:\tau}$ as

$$\begin{aligned}
\hat{y}_t &= g_\theta(h_t) = s(W_{hy} h_t + b_y) \quad \text{for } t \in \{1, \ldots \tau\} \\
h_t &= f_{\theta,t}(h_{t-1}) = a(W_{xh} D x_t + W_{hh} h_{t-1} + b_h) \quad \text{for } t \in \{1, \ldots \tau\},
\end{aligned}$$

where $s$ is, e.g., the soft-max function and $a$ is a nonlinear operation such as the hyperbolic tangent function and we fixed $h_0 = 0$.

To implement TP in this case, we consider making one step of gradient descent on the losses of the prediction, namely, computing

$$u_t = h_t - \gamma_h \partial_h \ell(y_t, g_\theta(h_t)) \quad \text{for } t \in \{1, \ldots, \tau\}.$$

Then we consider back-propagating the targets as

$$v_{t-1} = h_{t-1} + \partial_h f_\theta^\dagger(h_t)^\top (v_t - h_t + u_t - h_t) \quad \text{for } t \in \{\tau, \ldots, 1\}.$$

starting from $v_\tau = h_\tau$ such that if only $h_\tau$ was used to predict the output we retrieve the implementation of TP presented previously. The parameters of the network are then updated as presented in Sec. 2.

From a differentiable programming viewpoint, the computational scheme of our implementation of TP is the same as the one of gradient back-propagation once considering the displacements $u_t = u_t - h_t$ and $\lambda_t = v_t - h_t$, and replacing $\partial_h f_{\theta,t}(h_{t-1})$ by $\partial_h f_{\theta,t}^\dagger(h_t)^\top$ as presented in Fig. 10.

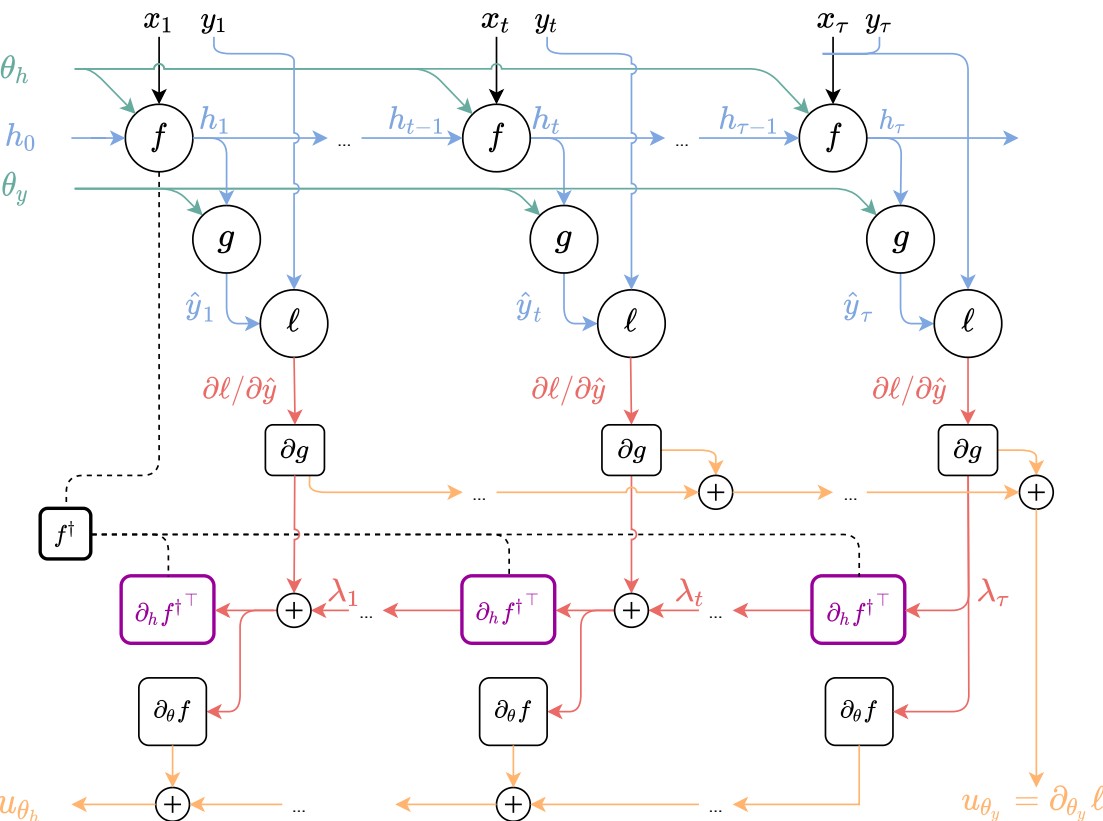

Fig. 10: Computational scheme of TP implemented for RNNs with intermediate outputs.

## C   Theoretical Insights

### C.1   Regularization and Noise Injection

In Section 2, we mentioned how the regularization used in the analytic formulation of the inverse layers could be interpreted as a counterpart of the noise injection heuristic used to learn parameterized inverses. We formalize this statement below. Recall that the transition functions take the form

$$f_{\theta,t}(h_t) = a(W_{xh}x_t + W_{hh}h_t + b_h). \tag{15}$$

Perturbed versions of this transition function read

$$f_{\theta+z,t}(h_t) = a((W_{xh} + Z_{xh})x_t + (W_{hh} + Z_{hh})h_{t-1} + b_h + z_h),$$

for $z \sim \mathcal{N}(0, \sigma^2\,\mathrm{I})$ decomposed into independent Gaussian random matrices $Z_{xh}, Z_{hh}$ and random vector $z_h$, all of which having independent Gaussian random entries with mean zero and variance $\sigma^2$.

**Lemma C.1.** *For $v_t \in a(\mathbb{R}^{d_h})$ and provided that the activation function $a$ is $\ell_a$-Lipschitz-continuous, the regularized inversion $f_{\theta,t}^{\dagger}(v_t) = (W_{hh}^{\top}W_{hh} + r\,\mathrm{I})^{-1}W_{hh}^{\top}(a^{-1}(v_t) - W_{xh}x_t - b_h)$ of $f_{\theta,t}$ defined in (15) minimizes an upper-bound on the variational formulation of the inverse subject to random perturbations, i.e., $\min_{v_{t-1}} \mathbb{E}_{z\sim\mathcal{N}(0,\sigma^2\,\mathrm{I})}\|f_{\theta+z,t}(v_{t-1}) - v_t\|_2^2$ for $\sigma^2 = r/d_h$.*

*Proof.* We have for $v_t \in a(\mathbb{R}^{d_h})$ and $a$, $\ell_a$-Lipschitz-continuous,

$$\|f_{\theta+z,t}(v_{t-1}) - v_t\|_2^2 = \mathbb{E}_{Z_{xh},Z_{hh},z_h}\|a\left((W_{xh} + Z_{xh})x_t + (W_{hh} + Z_{hh})v_{t-1} + b_h + z_h\right) - v_t\|_2^2$$
$$\leq \ell_a \mathbb{E}_{Z_{xh},Z_{hh},z_h}\|(W_{xh} + Z_{xh})x_t + (W_{hh} + Z_{hh})v_{t-1} + b_h + z_h - a^{-1}(v_t)\|_2^2,$$

The minimizer of the above upper bound is then

$$\hat{v}_{t-1} = \operatorname*{argmin}_{v_{t-1}\in\mathbb{R}^{d_h}} \mathbb{E}_{Z_{xh},Z_{hh},z_h}\|(W_{xh} + Z_{xh})x_t + (W_{hh} + Z_{hh})v_{t-1} + b_h + z_h - a^{-1}(v_t)\|_2^2$$

$$= \operatorname*{argmin}_{v_{t-1}\in\mathbb{R}^{d_h}} \mathbb{E}_{Z_{xh},Z_{hh},z_h}\Big[\frac{1}{2}v_{t-1}^{\top}(W_{hh} + Z_{hh})^{\top}(W_{hh} + Z_{hh})v_{t-1}$$

$$- v_{t-1}^{\top}(W_{hh} + Z_{hh})^{\top}(a^{-1}(v_t) - (W_{xh} + Z_{xh})x_t - b_h - z_h)\Big]$$

$$= \operatorname*{argmin}_{v_{t-1}\in\mathbb{R}^{d_h}} \left\{\frac{1}{2}v_{t-1}^{\top}(W_{hh}^{\top}W_{hh} + \mathbb{E}[Z_{hh}^{\top}Z_{hh}])v_{t-1} - v_{t-1}^{\top}W_{hh}^{\top}(a^{-1}(v_t) - W_{xh}x_t - b_h)\right\}$$

$$= (W_{hh}^{\top}W_{hh} + d_h\sigma^2\,\mathrm{I})^{-1}W_{hh}^{\top}(a^{-1}(v_t) - W_{xh}x_t - b_h) = f_{\theta,t}^{\dagger}(v_t),$$

for $f_{\theta,t}^{\dagger}(v_t)$ the regularized inversion used in our implementation with $r = d_h\sigma^2$.   □

If the activation function is the identity, the regularized inversion $f_{\theta,t}^{\dagger}$ is the exact minimizer of the variational formulation of the inverse subject to random perturbations. If the activation function is not the identity, we can still quantify the approximation used by using regularized inverses as shown below.

**Lemma C.2.** *Consider an injective and $\ell_a$-Lipschitz continuous activation function. For $v_t \in a(\mathbb{R}^{d_h})$, denote the objective of the variational formulation of the inverse subject to random perturbations as $F(v) = \mathbb{E}_{z\sim\mathcal{N}(0,\sigma^2\,\mathrm{I})}\|f_{\theta+z,t}(v) - v_t\|_2^2$ and denote $\hat{v} = f_{\theta,t}^{\dagger}(v_t)$ the approximate minimizer given by the regularized inversion for $r = d_h\sigma^2$. We have that*

$$F(\hat{v}) - \min_{v\in\mathbb{R}^{d_h}} F(v) \leq \left(1 - \frac{m_{a,\delta}}{\ell_a}\right)F(\hat{v}) + m_{a,\delta}\varepsilon_\delta,$$

*where, for $v^* \in \operatorname{argmin}_{v\in\mathbb{R}^{d_h}} F(v)$, denoting $\phi_{\theta,t}(v) = W_{xh}x_t + W_{hh}v + b_h$, we defined for any $\delta > 0$,*

$$m_{a,\delta} = \inf_{\substack{u=\phi_{\theta+z,t}(v^*),\|z\|_2\leq\delta \\ w=a^{-1}(v_t)}} \frac{\|a(u) - a(w)\|_2}{\|u - w\|_2}, \qquad \varepsilon_\delta = \int_{\|z\|_2\geq\delta}\|\phi_{\theta+z,t}(v^*) - a^{-1}(v_t)\|_2^2 e^{-\|z\|_2^2/2\sigma^2}dz.$$

*such that $m_{a,\delta} > 0$ since $a$ is injective and $\varepsilon_\delta$ vanishing for $\delta \to +\infty$.*

*Proof.* Decompose the transition function $f_{\theta,t}$ as $f_{\theta,t} = a \circ \phi_{\theta,t}$, where $\phi_{\theta,t}(v) = W_{xh}x_t + W_{hh}v + b_h$ and $\phi_{\theta+z,t}(v) = (W_{xh} + Z_{xh})x_t + (W_{hh} + Z_{hh})h_{t-1} + b_h + z_h$. Denote $v^* \in \arg\min_{v \in \mathbb{R}^{d_h}} F(v)$. We have

$$
\begin{aligned}
F(v^*) &= \int_{\|z\|_2 \leq \delta} \|a(\phi_{\theta+z,t}(v^*)) - v_t\|_2^2 e^{-\|z\|_2^2/2\sigma^2} dz + \int_{\|z\|_2 \geq \delta} \|a(\phi_{\theta+z,t}(v^*)) - v_t\|_2^2 e^{-\|z\|_2^2/2\sigma^2} dz \\
&\geq \int_{\|z\|_2 \leq \delta} \|a(\phi_{\theta+z,t}(v^*)) - v_t\|_2^2 e^{-\|z\|_2^2/2\sigma^2} dz \\
&\geq \int_{\|z\|_2 \leq \delta} m_{a,\delta} \|\phi_{\theta+z,t}(v^*) - a^{-1}(v_t)\|_2^2 e^{-\|z\|_2^2/2\sigma^2} dz \\
&\geq m_{a,\delta} \int \|\phi_{\theta+z,t}(v^*) - a^{-1}(v_t)\|_2^2 e^{-\|z\|_2^2/2\sigma^2} dz - m_{a,\delta}\varepsilon_\delta \\
&\overset{(i)}{\geq} m_{a,\delta} \int \|\phi_{\theta+z,t}(\hat{v}) - a^{-1}(v_t)\|_2^2 e^{-\|z\|_2^2/2\sigma^2} dz - m_{a,\delta}\varepsilon_\delta \\
&\geq m_{a,\delta} \ell_a^{-1} F(\hat{v}) - m_{a,\delta}\varepsilon_\delta,
\end{aligned}
$$

where in $(i)$ we used that $\hat{v} = \arg\min_v \mathbb{E}_z \|\phi_{\theta+z,t}(v) - a^{-1}(v_t)\|_2^2$ and we defined

$$
m_{a,\delta} = \inf_{\substack{u = \phi_{\theta+z,t}(v^*), \|z\|_2 \leq \delta \\ w = a^{-1}(v_t)}} \frac{\|a(u) - a(w)\|_2}{\|u - w\|_2}, \qquad \varepsilon_\delta = \int_{\|z\|_2 \geq \delta} \|\phi_{\theta+z,t}(v^*) - a^{-1}(v_t)\|_2^2 e^{-\|z\|_2^2/2\sigma^2} dz.
$$

$\square$

Lemma C.2 shows that if the solution of the variational formulation of the inverse subject to random perturbations is in a region of the activation function where the activation function is nearly the identity, i.e., such that $m_{a,\delta} \approx 1$ for, e.g., $\delta = 3\sigma$ such that $\varepsilon_\delta \ll 1$, then the regularized inversion of the transition function we propose nearly minimizes the objective of the variational formulation of the inverse subject to random perturbations.

## C.2 Gradient Back-Propagation vs Target Propagation

**Lemma 3.1.** *The difference between the oracle returned by gradient back-propagation $\partial_{\theta_h} \ell(y, F_\theta(x_{1:\tau}))$ and the oracle returned by target propagation $u_{\theta_h}$ can be bounded as*

$$
\|\partial_{\theta_h} \ell(y, F_\theta(x_{1:\tau})) - u_{\theta_h}\| \leq \sum_{t=1}^{\tau} c_t \|\partial_{\theta_h} f_{\theta,t}(h_{t-1})\| \|\partial_{h_\tau} \ell(y, g_\theta(h_\tau))\| \sup_{t=1,\dots,\tau} \|\partial_{h_{t-1}} f_{\theta,t}(h_{t-1}) - \partial_{h_t} f_{\theta,t}^\dagger(h_t)^\top\|,
$$

*where $c_t = \sum_{s=0}^{t-1} a^s b^{t-1-s}$ with $a = \sup_{t=1,\dots\tau} \|\partial_{h_{t-1}} f_{\theta,t}(h_{t-1})\|, b = \sup_{t=1,\dots\tau} \|\partial_{h_t} f_{\theta,t}^\dagger(h_t)^\top\|$.*

*For regularized inverses, we have, denoting $z_t = W_{xh}x_t + W_{hh}h_{t-1} + b_h$,*

$$
\|\partial_{h_{t-1}} f_{\theta,t}(h_{t-1}) - \partial_{h_t} f_{\theta,t}^\dagger(h_t)^\top\| \leq \|W_{hh}^\top\| \left( \|\nabla a(z_t) - \nabla a(z_t)^{-1}\| + \|I - (W_{hh}^\top W_{hh} + r\,I)^{-1}\| \|\nabla a(z_t)^{-1}\| \right).
$$

*Proof.* The first claim is a direct application of Lemma C.3 and the second claim follows from the formulation of the regularized inverse, using that $\nabla a^{-1}(h_t) = \nabla a(a^{-1}(h_t))^{-1} = \nabla a(z_t)^{-1}$. $\square$

**Lemma C.3.** *Given $A_1, \dots, A_n, B_1, \dots, B_n \in \mathbb{R}^{p \times p}$, for any sub-multiplicative matrix norm $\|\cdot\|$, and any $1 \leq t \leq n$,*

$$
\left\| \prod_{i=1}^{t} A_i - \prod_{i=1}^{t} B_i \right\| \leq \delta \sum_{i=0}^{t-1} a^i b^{t-1-i},
$$

*where $a = \sup_{i=1,\dots,n} \|A_i\|$, $b = \sup_{i=1,\dots n} \|B_i\|$ and $\delta = \sup_{i=1,\dots,n} \|A_i - B_i\|$.*

*Proof.* Define for $t \geq 1$, $\delta_t = \|\prod_{i=1}^{t} A_i - \prod_{i=1}^{t} B_i\|$, we have

$$\delta_t \leq \left\| A_t \left( \prod_{i=1}^{t-1} A_i - \prod_{i=1}^{t-1} B_i \right) + (A_t - B_t) \prod_{i=1}^{t-1} B_i \right\| \leq a\delta_{t-1} + \delta b^{t-1} \leq \delta \sum_{i=0}^{t-1} a^i b^{t-1-i}.$$

$\square$

A convergence to a stationary point for TP can be derived from classical results on an approximate gradient descent detailed below (the proof is akin to the results of Devolder et al. (2014)).

**Corollary C.4** (Corollary of Lemma C.5)**.** *Denote $\varepsilon_k$ a bound on the difference between the oracle returned by gradient back-propagation and by target-propagation both applied to the whole dataset. Provided that the objective is $L$-smooth and the stepsizes of TP are chosen such that $\gamma = \gamma_h \gamma_y < 1/L$, after $K$ iterations, we get*

$$\min_{k \in \{0,\ldots,K-1\}} \|\nabla \mathcal{L}(\theta^{(k)})\|_2^2 \leq \frac{2 \left( \mathcal{L}(\theta^{(0)}) - \min_{\theta \in \mathbb{R}^d} \mathcal{L}(\theta) \right)}{\gamma K}) + \frac{1}{K} \sum_{k=0}^{K-1} \varepsilon_k^2.$$

*where $\mathcal{L}(\theta) = \frac{1}{n} \sum_{i=1}^{n} \ell(F_\theta(x_{1:\tau,i}), y_i)$ with $F_\theta(x_{1:\tau,i})$ the output of the RNN on a sample $(x_{1:\tau,i}, y_i)$ and $\ell$ the chosen loss.*

**Lemma C.5.** *Let $f : \mathbb{R}^p \to \mathbb{R}$ be an $L$-smooth function. Consider an approximate gradient descent on $f$ with step size $0 \leq \gamma \leq 1/L$, i.e., $\theta^{(k+1)} = \theta^{(k)} - \gamma \widehat{\nabla} f(\theta^{(k)})$, where $\|\widehat{\nabla} f(\theta^{(k)}) - \nabla f(\theta^{(k)})\|_2 \leq \varepsilon_k$. After $K$ iterations, this method satisfies,*

$$\min_{k \in \{0,\ldots,K-1\}} \|\nabla f(\theta^{(k)})\|_2^2 \leq \frac{2(f(\theta^{(0)}) - \min_{\theta \in \mathbb{R}^p} f(\theta))}{\gamma K} + \frac{1}{K} \sum_{k=0}^{K-1} \varepsilon_k^2.$$

*Proof.* Denote $g^{(k)} = \widehat{\nabla} f(\theta^{(k)}) - \nabla f(\theta^{(k)})$ for all $k \geq 0$. By $L$-smoothness of the objective, the iterations of the approximate gradient descent satisfy, using in $(i)$ that $L\gamma \leq 1$,

$$f(\theta^{(k+1)}) \leq f(\theta^{(k)}) + \nabla f(\theta^{(k)})^\top (\theta^{(k+1)} - \theta^{(k)}) + \frac{L}{2} \|\theta^{(k+1)} - \theta^{(k)}\|_2^2$$

$$= f(\theta^{(k)}) - \gamma \|\nabla f(\theta^{(k)})\|_2^2 - \gamma \nabla f(\theta^{(k)})^\top g^{(k)} + \frac{L\gamma^2}{2} \|\nabla f(\theta^{(k)}) + g^{(k)}\|_2^2$$

$$= f(\theta^{(k)}) - \gamma \left( 1 - \frac{L\gamma}{2} \right) \|\nabla f(\theta^{(k)})\|_2^2 + \frac{L\gamma^2}{2} \|g^{(k)}\|_2^2 + \gamma(L\gamma - 1)\nabla f(\theta^{(k)})^\top g^{(k)}$$

$$\overset{(i)}{\leq} f(\theta^{(k)}) - \gamma \left( 1 - \frac{L\gamma}{2} \right) \|\nabla f(\theta^{(k)})\|_2^2 + \frac{L\gamma^2}{2} \|g^{(k)}\|_2^2 + \gamma(1 - L\gamma)\|\nabla f(\theta^{(k)})\|_2 \|g^{(k)}\|_2$$

$$\leq f(\theta^{(k)}) - \gamma \left( 1 - \frac{L\gamma}{2} \right) \|\nabla f(\theta^{(k)})\|_2^2 + \frac{L\gamma^2}{2} \|g^{(k)}\|_2^2 + \frac{\gamma(1 - L\gamma)}{2} (\|\nabla f(\theta^{(k)})\|_2^2 + \|g^{(k)}\|_2^2)$$

$$\leq f(\theta^{(k)}) - \frac{\gamma}{2} \|\nabla f(\theta^{(k)})\|_2^2 + \frac{\gamma}{2} \|g^{(k)}\|_2^2.$$

Summing from $k = 0$ to $K - 1$ and rearranging the terms, we get

$$\sum_{k=0}^{K-1} \|\nabla f(\theta^{(k)})\|_2^2 \leq \frac{2(f(\theta^{(0)}) - \min_{\theta \in \mathbb{R}^p} f(\theta))}{\gamma} + \sum_{k=0}^{K-1} \varepsilon_k^2.$$

Taking the minimum of $\|\nabla f(\theta^{(k)})\|_2^2$ and dividing by $K$ we get the result.

$\square$

### C.3 Target Propagation vs Gauss-Newton updates

We discuss the interpretation of Target Propagation (TP) as a Gauss-Newton (GN) method which was proposed by Bengio (2020) and Meulemans et al. (2020). As already mentioned in Sec. 3, the main similarity between TP and GN is the fact that both TP and GN use the inverse of the gradients or approximations thereof. In this section, we shall discuss this interpretation for feed-forward networks to follow the claims of Meulemans et al. (2020). Namely, we consider here a network defined by $L$ weights $W_1, \ldots, W_L$ and $L$ activation functions $a_1, \ldots, a_L$ which transform an input $x_0$ into an output $x_L$ by computing (no biases were considered by Meulemans et al. (2020)),

$$x_t = f_t(x_{t-1}) = a_t(W_t x_{t-1}) \quad \text{for } t \in \{1, \ldots, L\}$$

Denoting $\phi(x, \theta)$ the output of the network for an input $x = x_0$, with $\theta = (W_1, \ldots, W_L)$ being the parameters of the network, the objective consists in minimizing the loss between the outputs of the network and the sample outputs, i.e., minimizing $\mathcal{L}(y, \phi(x, \theta))$ for pairs of input-output samples $(x, y)$.

**GN step.** Recall first the rationale of a GN step for such feed-forward networks with a squared-loss, which amount to solving

$$\min_{\theta \in \mathbb{R}^p} \frac{1}{n} \sum_{i=1}^{n} \|\phi(x_i, \theta) - y_i\|_2^2.$$

A GN step amounts to linearize the non-linear function $\phi$ around a current set of parameters $\theta^{(k)}$ and solve the corresponding least-square problems to define the next set of parameters, i.e,

$$\theta^{(k+1)} = \operatorname*{argmin}_{\theta} \frac{1}{n} \sum_{i=1}^{n} \|\phi(x_i, \theta^{(k)}) + \partial_\theta \phi(x_i, \theta^{(k)})^\top (\theta - \theta^{(k)}) - y_i\|_2^2$$

$$= \theta^{(k)} - \left( \sum_{i=1}^{n} \partial_\theta \phi(x_i, \theta^{(k)}) \partial_\theta \phi(x_i, \theta^{(k)})^\top \right)^{-1} \left( \sum_{i=1}^{n} \partial_\theta \phi(x_i, \theta^{(k)}) \left( \phi(x_i, \theta^{(k)}) - y_i \right) \right).$$

To consider TP as an approximate GN method we need the following considerations.

1. Consider the iteration on a mini-batch of size 1, s.t.

$$\theta^{(k+1)} = \theta^{(k)} - \left( \partial_\theta \phi(x_i, \theta^{(k)}) \partial_\theta \phi(x_i, \theta^{(k)})^\top \right)^{-1} \left( \partial_\theta \phi(x_i, \theta^{(k)}) \left( \phi(x_i, \theta^{(k)}) - y_i \right) \right).$$

2. Consider that the gradients of the networks are invertible, s.t.

$$\theta^{(k+1)} = \theta^{(k)} - \left( \partial_\theta \phi(x_i, \theta^{(k)}) \right)^{-\top} \left( \phi(x_i, \theta^{(k)}) - y_i \right).$$

3. Consider updating only one set of parameters $\theta_l = W_l$ , s.t.,

$$\theta_l^{(k+1)} = \theta_l^{(k)} - \left( \partial_{\theta_l} \phi(x_i, \theta^{(k)}) \right)^{-\top} \left( \phi(x_i, \theta^{(k)}) - y_i \right).$$

with

$$\partial_{\theta_l} \phi(x_i, \theta^{(k)}) = \partial_{\theta_l} f_l(x_{l-1}) \partial_x f_{l+1}(x_l) \ldots \partial_x f_L(x_{L-1})$$

so that, provided that all matrices inside the matrix multiplication are invertible, we get

$$\partial_{\theta_l} \phi(x_i, \theta^{(k)})^{-T} = \partial_{\theta_l} f_l(x_{l-1})^{-T} \partial_x f_{l+1}(x_l)^{-T} \ldots \partial_x f_L(x_{L-1})^{-T}$$

4. Finally, ignore the last inversion and replace it by the gradient on the parameters $\theta_l$, then we get an iteration similar to TP, with

$$\theta_l^{(k+1)} = \theta_l^{(k)} - \partial_{\theta_l} f_l(x_{l-1}) \partial_x f_{l+1}(x_l)^{-T} \ldots \partial_x f_L(x_{L-1})^{-T} \partial_{x_L} \mathcal{L}(y, x_L)$$

for $\mathcal{L}$ a squared loss. Namely, we keep the inversion of the gradients of the intermediate functions.

Our objective here is to question whether viewing TP as a GN step with the approximations explained above is meaningful or not.

**Does the original TP formulation approximate GN?** Meulemans et al. (2020) start by considering the original TP formulation, i.e., targets computed as $v_t = \psi_t(v_{t+1})$ for $\psi_t$ an approximate inverse of $f_t$ and with $v_L = x_L - \eta \partial_{x_L} \ell(y, x_L)$. Meulemans et al. (2020, Lemma 1) show then that, provided that we use the exact inverse, $\psi_t = f_t^{-1}$,

$$\Delta x_t = v_t - x_t = -\eta \prod_{s=t}^{L-1} \partial_{x_s} f_{s+1}(x_s)^{-\top} \partial_{x_L} \ell(y, x_L) + O(\eta^2).$$

(Meulemans et al., 2020, Theorem 2) conclude that (i) for mini-batches of size 1, (ii) for a squared loss, (iii) for invertible $f_t$, as $\eta \to 0$, TP uses a Gauss-Newton optimization with block diagonal approximation to compute the targets in the sense that as $\eta \to 0$,

$$\Delta x_t \approx -\eta \partial_{x_t} (f_{t+1} \circ \ldots \circ f_L)^{-\top}(x_t).$$

As the stepsize of any optimization algorithm tends to 0, they all are the same, since the update would be 0 in all cases. To make the claim of Meulemans et al. (2020) more precise, the constants hidden in $O(\eta^2)$ need to be detailed in order to understand in which regimes of the stepsize the approximation is meaningful.

Assuming the inverses $\psi_t$ to be $\ell_\psi$ Lipschitz continuous and $L_\psi$-smooth (i.e. with $L_\psi$-Lipschitz continuous gradients), a quick look at the proof of Lemma 1 of Meulemans et al. (2020) shows that

$$v_t - x_t = -\eta \prod_{s=t}^{L-1} \partial_{x_s} f_{s+1}(x_s)^{-\top} \partial_{x_L} \ell(y, x_L) + \xi_t$$

$$\|\xi_t\|_2 \leq \delta_t$$

$$\delta_s \leq L_\psi \delta_{s+1}^2 + \ell_\psi \delta_{s+1} + L_\psi \ell_\psi^2 \eta^2 \|\partial_{x_L} \ell(y, x_L)\|_2^2 \quad \text{for } s \in \{t, \ldots, L-1\}$$

$$\delta_L \leq \frac{L_\psi}{2} \eta^2 \|\partial_{x_L} \ell(y, x_L)\|_2^2.$$

Hence the approximation error is then of the order of $\|\xi_t\|_2 \leq L_\psi \|\partial_{x_L} \ell(y, x_L)\|_2^2 (\sum_{s=0}^{L-t} \ell_\psi^s) \eta^2 + o(\eta^2)$, that is the approximation error may be valid for $\eta \leq (L_\psi \|\partial_{x_L} \ell(y, x_L)\|_2^2 (\sum_{s=0}^{L-t} \ell_\psi^s))^{-1/2}$. Yet, in practice, TP does not appear to use very small stepsizes. Moreover, if the similarity of TP with GN could explain its efficiency, then by the reasoning of Meulemans et al. (2020), the original TP formulation should be efficient. Yet, the original TP formulation has never been shown to produce satisfying results.

**Does TP with the difference target propagation approximate GN?** Meulemans et al. (2020) make a similar claim for TP with the Difference Target Propagation formula, i.e., $v_t = x_t + \psi_t(v_{t+1}) - \psi_t(x_{t+1})$. Namely, Meulemans et al. (2020, Lemma 3) show that

$$\Delta x_t = v_t - x_t = -\eta \prod_{s=t}^{L-1} \partial_{x_s} \psi_s(x_s)^\top \partial_{x_L} \ell(y, x_L) + O(\eta^2).$$

Once again, for the claim to be meaningful beyond infinitesimal stepsizes, the terms in $O(\eta^2)$ need to be detailed. If we use a linearized version of the difference target propagation formula as presented in (2), namely $v_t - x_t = \partial_{x_{t+1}} \psi_t(x_{t+1})^\top (v_{t+1} - x_{t+1})$ , then we have the *equality*

$$\Delta x_t = v_t - x_t = -\eta \prod_{s=t}^{L-1} \partial_{x_s} \psi_s(x_s)^\top \partial_{x_L} \ell(y, x_L)$$

and the idea that TP could be seen as an approximate GN method may be pursued in a meaningful way. However the error of approximation of the inverse of the gradients must be taken into account in order to understand the validity of the approach.

**Propagating the approximation error of the gradient inverses.** We compute the approximation error incurred by composing gradients of the inverse instead of inverses of gradients. Formally, the approximation error for one layer can be estimated under the assumption that

$$\psi_t(f_t(x_{t-1})) = x_{t-1} + e(x_{t-1}), \tag{16}$$

with $e$ an $\varepsilon$-Lipschitz continuous function and the assumption that the minimal singular value $\sigma$ of $\partial_{x_t} f_t(x_{t-1})$ is positive.

The function $e$ a priori depends on the parameters of the layer; we ignore this dependency and simply consider $e$ to be $\varepsilon$-Lipschitz continuous for all $\theta$. For a function $e$, its Lipschitz continuity constant is $\varepsilon = \sup_x \sup_{\|\lambda\|_2 \leq 1} \|\partial_x e(x)^\top \lambda\|_2 = \sup_x \|\partial_x e(x)\|$, where $\|\cdot\|$ denotes the spectral norm. By differentiating both sides of Eq. (16), we get

$$\partial_x f_t(x_{t-1}) \partial_x \psi_t(x_t) = \mathrm{I} + \partial_x e(x_{t-1}).$$

By assuming the minimal singular value $\sigma$ of $\partial_x f_t(x_{t-1})$ to be positive, we get that $\partial_x f_t(x_{t-1})$ is invertible and so

$$\partial_x \psi_t(x_t) = \partial_x f_t(x_{t-1})^{-1}(\mathrm{I} + \partial_x e(x_{t-1})).$$

Hence $\partial_x \psi_t(x_t)$ is $\sigma^{-1}(1 + \varepsilon)$ Lipschitz-continuous and

$$\| (\partial_x f_t(x_{t-1}))^{-1} - \partial_x \psi_t(x_t) \| \leq \frac{\varepsilon}{\sigma}. \tag{17}$$

Now for multiple compositions, using Lemma C.3, we get

$$\| (\partial_x f_1(x_0))^{-1} \ldots (\partial_x f_L(x_{L-1}))^{-1} - \partial_x \psi_1(x_1) \ldots \partial_x \psi_L(x_L) \| \leq \frac{(1+\varepsilon)^L}{\sigma^L}.$$

Therefore the accumulation error diverges with the length $L$ of the network as soon as $\varepsilon \geq \sigma - 1$.

**Testing the hypothesis that TP could be interpreted as using GN updates directions.** Here we come back to the setting of RNNs presented in the paper. In this case the length of the compositions of layers is $\tau$ and according to the previous discussion, the error of approximation of the product of the inverse of the gradients by the product of the gradients of the approximate inverses could easily diverge as $\tau$ grows (long sequences). Nevertheless, by using analytical formulas for the inverses, we can ensure that the approximation error is zero, which would correspond then to the ideal setting where TP uses GN update directions for the hidden states.

Formally, in the context of RNNs, a Gauss-Newton update direction for the hidden states is given as (ignoring the inverse of the output function)

$$-\gamma_h \prod_{s=t+1}^{\tau-1} (\partial_h f_{t+1,\theta}(h_t))^{-\top} \partial_h \ell(y, g_\theta(h_\tau)),$$

If no regularization is used in the definition of the regularized inverse, i.e., if we use

$$f_{\theta,t}^{-1}(h_t) = (W_{hh}^\top W_{hh})^{-1} W_{hh}^\top (a^{-1}(h_t) - W_{xh}x_t - b_h),$$

which requires the inverse of $W_{hh}$ to be well defined, we would get

$$\partial f_{\theta,t}^{-1}(h_t) = \partial_h f_{t+1,\theta}(h_t)^{-1}.$$

The updates of TP using the formula (2) would then be exactly the ones of a GN on the hidden states as presented by Meulemans et al. (2020), i.e.,

$$v_t - h_t = -\gamma_h \prod_{s=t+1}^{\tau-1} (\partial_h f_{t+1,\theta}(h_t))^{-\top} \partial_h \ell(y, g_\theta(h_\tau)).$$

So by considering our implementation without regularization, we can test whether the interpretation of TP as an approximate GN method is meaningful in terms of optimization convergence. As shown in Fig. 7, it appears that regularizing the inverses is necessary to obtain convergence, hence the interpretation of TP as GN may not be sufficient to explain why TP can converge.

|  | BP | | TP | |
| --- | --- | --- | --- | --- |
|  | $\gamma$ | $\gamma_h$ | $\gamma_\theta$ | $\kappa$ |
| Temporal order problem length 60 | $10^{-5}$ | $10^{-2}$ | $10^{-1}$ | 10 |
| Temporal order problem length 120 | $10^{-5}$ | $10^{-2}$ | $10^{-2}$ | 1 |
| Adding problem | $10^{-3}$ | $10^{-1}$ | $10^{-1}$ | 1 |
| MNIST pixel by pixel | $10^{-6}$ | $10^{-4}$ | $10^{-1}$ | 1 |
| MNIST pixel by pixel permuted | $10^{-4}$ | $10^{-4}$ | $10^{-1}$ | 1 |
| CIFAR | $10^{-3}$ | $10^{-2}$ | $10^{-2}$ | 10 |
| FashionMNIST with GRU | $10^{-2}$ | $10^{-1}$ | $10^{-2}$ | 1 |
| Penn Treebank with RNN | $10^{-2}$ | $10^{-3}$ | $10^{-1}$ | 1 |

Table 1: Hyper-parameters chosen for Fig. 3 and 4.

# D   Experimental Details

## D.1   Initialization and Hyper-Parameters

**Initialization and data generation.**   In all experiments, the weights of the RNN are initialized as random orthogonal matrices, and the biases are initialized as 0 as presented by Le et al. (2015) and Manchev & Spratling (2020). For all experiments, the data was not normalized, as done by Manchev & Spratling (2020). We kept a setting as similar as possible as the one of Manchev & Spratling (2020) to be able to compare target propagation with regularized or parameterized inverses.

**Hyper-parameters.**   In the synthetic tasks, for BP we used a momentum of 0.9 with Nesterov accelerated gradient scheme as done by Manchev & Spratling (2020). Otherwise, we did not use any momentum for the experiment on MNIST pixel by pixel presented in the main paper. The learning rates of BP and the parameters of TP were found by a grid-search on a $\log_{10}$ basis and are presented in Table 1. We did not add a regularization term in the training of the RNNs.

For Fig. 7, we used batch sizes of size 512 and performed a grid search for the stepsizes of BP and for the stepsizes $\gamma_h$ of TP while keeping the same regularization $r$ and stepsize $\gamma_\theta$ to the parameters found for the length 784.

**Software.**   We used Python 3.8 and PyTorch 1.6. The RNN was coded using the cuDNN implementation available in PyTorch that is highly optimized for computing forward-backward passes on RNNs for gradient back-propagation.

**Hardware.**   All experiments were performed on GPUs using Nvidia GeForce GTX 1080 Ti (12G memory). Each experiment only used one gpu at a time (clock speed 1.5 Ghz).

**Time evaluation.**   On our GPU, we observed that for the MNIST pixel by pixel experiment, 200 iterations (each iteration considering 16 samples) were taking approximately 60s for BP and 800s for TP. Note that with larger batch-sizes the cost of the regularized inversion would be amortized by the fact that more samples are treated simultaneously. We kept the setting of Manchev & Spratling (2020) for ease of comparison.

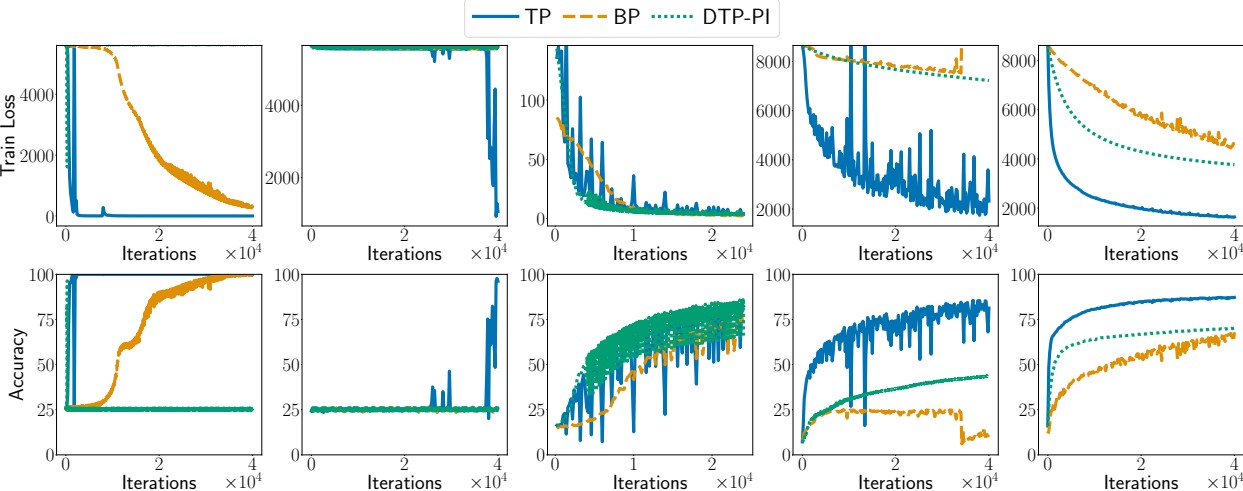

Fig. 11: Comparison of our implementation of TP denoted TP, against TP with a Difference Target Propagation formula and Parameterized Inverses, denoted DTP-PI and gradient Back-Propagation, denoted BP. From Left to right: Temporal order task with $T = 60$, Temporal order task with $T = 120$, Adding problem with $T = 30$, MNIST pixel by pixel and MNIST pixel by pixel with permuted images.

## D.2 Additional Experiments

**Comparison of TP with regularized or parameterized inverses.** In Fig. 11, we evaluate the performance of our implementation of TP using regularized inverses and linearized propagation as opposed to the implementation of Manchev & Spratling (2020) using the difference target propagation formula and parameterized inverses on all datasets studied by Manchev & Spratling (2020) using the reported hyper-parameters chosen by Manchev & Spratling (2020). We observe that our implementation generally outperforms the one of Manchev & Spratling (2020) except for the addition task where they perform on-par. As observed in Fig. 5, the difference of performance can be explained by the use of regularized inverses while the linearized formulation used in our implementation has the advantage to easily adapt TP for other architectures.

**Visualization of norms of targets.** On Fig. 12, we observe that the norm of the displacements $\lambda_t = v_t - h_t$ defined in Sec. 3 propagated by TP do not necessarily enjoy more stable norms than the gradients $\partial\ell/\partial h_t$ propagated by BP as explained in Appendix A for $t$ varying along the length of the sequence. This phenomenon is in contrast with the better performance of TP over BP for this task, namely, predicting images from MNIST with RNNs and suggests interesting avenues for future research.

**Gradient norms and spectral radius analyses.** On Fig. 13, we present, as done by Manchev & Spratling (2020, Figure 3) for TP with parameterized inverses, the evolution of (i) the norm of the oracle directions computed by either TP or BP for all transition parameters $\theta_h$, (ii) the spectral radius of the transition matrix $W_{hh}$, on the MNIST pixel by pixel experiment. We observe that TP provides smaller oracle direction norms which allowed for larger stepsizes with slightly less variance of the gradients. On the other hand the spectral radius of the transition matrix appears to grow for both oracles with TP having a larger growth at the start.

**TP vs BP in terms of time.** To account for the additional cost of inversion for each mini-batch, we consider the convergence of the algorithms in time rather than in iterations. We found that, on average, 1 iteration of BP takes approximately 13 times less time than one iteration of TP in our implementation (note that BP benefits from highly optimized implementations for GPU machines, and TP could potentially also benefit from the same optimized implementations). Therefore we ran BP for 13 times more iterations than TP and multiplied the number of iterations by the approximate time needed for each iteration for all

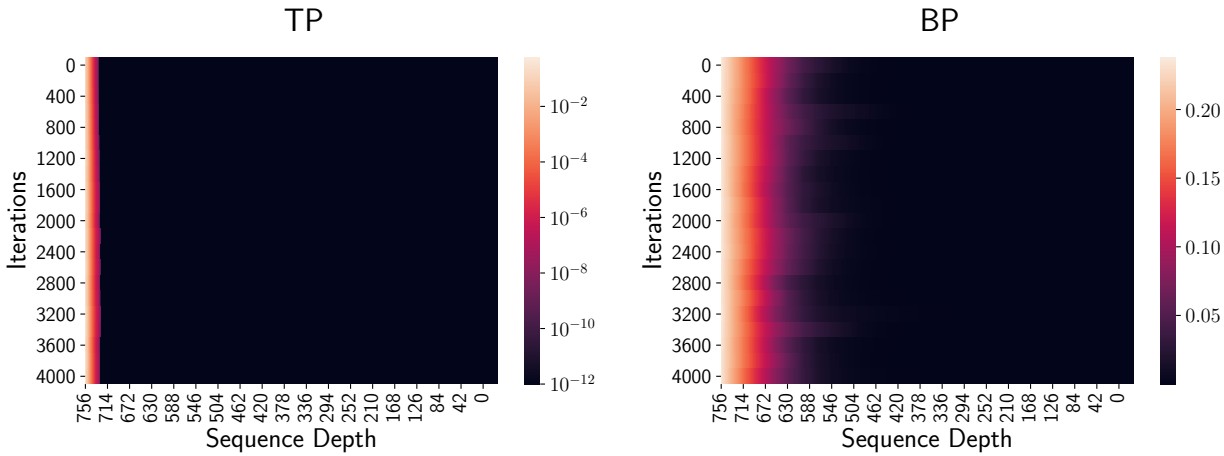

Fig. 12: Norms of displacements or gradients along the sequence of pixels analyzed by an RNN to predict MNIST digits.

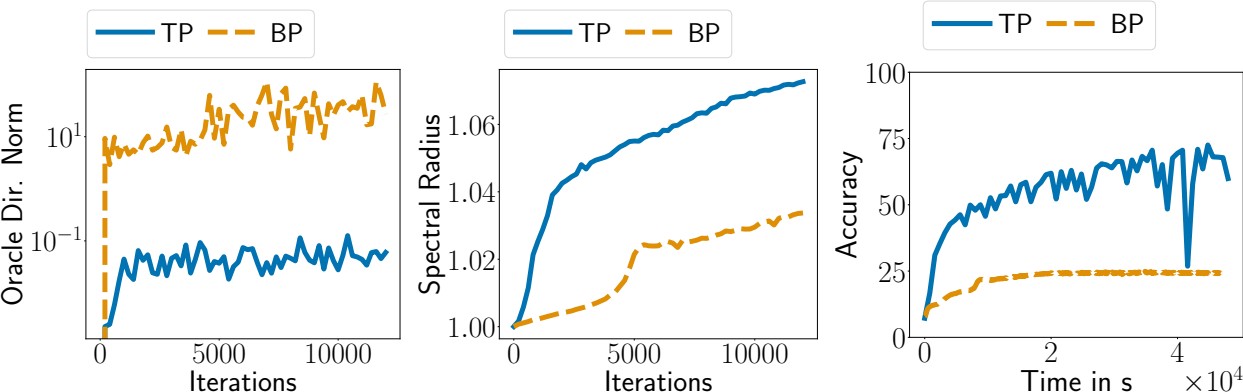

Fig. 13: Norms of oracle directions and spectral radii along iterations. Fig. 14: MNIST pixel by pixel in time.

algorithms. In Fig. 14, we observe that in time too, TP performs better than BP, which stays stuck at an accuracy of approximately 22%.

