# OpenReview forum: "Target Propagation via Regularized Inversion for Recurrent Neural Networks"
_TMLR — Accepted by TMLR_

### Review · Reviewer_wpQC · 2022-08-16

**Summary Of Contributions:**

In the paper under review, the authors present a simple version of target propagation based on regularized inversions of neural network layers that sheds light on the relevance of target propagation from an optimization viewpoint, and the proposed target propagation using the analytical formulation, and the proposed target propagation is simple to implement using differentiable programming. The authors verified the efficiency of target propagation in training recurrent neural networks.  Moreover, the authors consider the regimes in which the computational complexity of target propagation can outperform backpropagation.

**Requested Changes:**

Below are some details comments.

From Figure 1, I cannot see the computational advantages of target propagation over gradient back-propagation. It seems to me that for target propagation, we also need to store the intermediate states in the forward propagation. Moreover, I wonder about the numerical stability of computing the gradient of the inverse of the function.


The experiments are rather simple. I would suggest authors validate them on a few more recurrent neural network benchmarks, e.g., TIMIT and character- and word-level Penn treebank tasks.


I would suggest the authors do some visualization of overcoming vanishing and exploding gradient issues using target propagation. Perhaps the authors can mimic what did in Figure 2 in the paper https://proceedings.neurips.cc/paper/2020/file/149ef6419512be56a93169cd5e6fa8fd-Paper.pdf



**Strengths And Weaknesses:**

Overall, the paper is well-written, and I am happy to recommend this paper for publication after major revision.

---

> ### Author Response · Authors · 2022-09-03
> **Answer to Reviewer wpQC**
>
> Thank you for your detailed and insightful comments! Below we detail the changes we have done in our manuscript (highlighted in teal) following your suggestions, and we answer your comments.
> - We are not sure to understand the comment on the computational advantages of TP. Our objective here is to integrate the idea of target propagation in a differentiable framework to evaluate each of its heuristics from an optimization viewpoint on several experiments, not necessarily to show that TP is better than BP in all cases. For example, Fig. 7 presents regimes where BP is better than TP. In terms of computational efficiency, for very long sequences, we showed in Fig. 13 that the overhead given by computing the inverse of the transition function can be amortized by the efficiency of TP, an insight justified by the analysis of the computational cost of TP done in Sec. 3. On a memory cost standpoint our implementation does not offer benefits, but it informs how to parameterize for example implementing TP using parameterized inverses.
> - In Fig. 4, last panel, we added a comparison of TP against BP on the task of predicting words on the Penn Treebank dataset, which shows that our implementation can easily be adapted to this setting while optimizing the training loss correctly in this task.
> - Thank you for your suggestion about vanishing/exploding gradients and the relevant reference. We present in Fig. 12 in Appendix D.2 the norm of the displacements along the sequence analyzed by an RNN and compare it to its counterpart when using BP. Interestingly, TP does not necessarily lead to a better propagation of the displacements that can be seen as a counterpart of the gradients computed by BP. This additional experiment gives more insights about TP and helps to understand its advantages/disadvantages in contrast with the pure performance plot given in Fig. 4.
>
> We thank you again for your insightful comments that helped us to give a better perspective on TP and to improve our manuscript.

---

### Review · Reviewer_9aYY · 2022-08-20

**Summary Of Contributions:**

This paper proposes a variant of the target propagation for training recurrent neural networks. The method builds on the work [Lee et al. (2015)], [Bengio (2020)], and so on. The key differences from the previous methods are to utilize (i) linear approximation eq. (2) of the inversion instead of the difference target propagation and (ii) (regularized) analytic inversion instead of optimizing the parameterized one.
This paper shows the competitive computational efficiency with the back-propagation, shows several aspects of the method (e.g., connection with the noise injection, linearized inversion, and Gauss-Newton method), and estimates the gap between the proposed method and the back-propagation (Lemma 3.1). Moreover, the better performance than the back-propagation is empirically verified on synthetic datasets and image datasets.

**Broader Impact Concerns:**

Not applicable.

**Requested Changes:**

- It would be nice if the authors could show the convergence more rigorously. If the comparable convergence guarantee with the back-propagation can be guaranteed, the use of this method can also be justified.
- Can you conduct experiments on some NLP datasets? Moreover, the proposed method should be empirically compared with the existing target propagation methods on the other datasets other than that in Appendix.


**Strengths And Weaknesses:**

The method's primary advantages over the previous target propagation are the computational efficiency and estimation accuracy of the inversion map, which are attributed to regularized analytic inversion. This method involves the computation of inverse matrix, but computation only needs to be done once per sequence, and its accuracy is also guaranteed to some extent. On the other hand, the existing methods require the estimation of the inverse at each time step, and there is no guarantee for accuracy. This improvement (especially the computational cost) over the existing target propagation would be nice.

On the other hand, the novelty and contribution seem to be low, as summarized below.

- The computational efficiency of the proposed method and back-propagation should be evaluated regarding the complexity to achieve the same accuracy. If there is a performance difference between each iteration of these methods, then a comparison of per-iteration costs in Section 3 is not very interesting.
- The aim of Lemma 3.1, which bounds the difference between target propagation and back-propagation, is unclear. If this gap should be small, then I am not sure why the proposed method is used.
In Appendix, this lemma is used to discuss the convergence of the proposed method, but the obtained bound seems vacuous because the difference between these two steps, which should be smaller than a given threshold $\epsilon$, is not properly evaluated.
- Given [Bengio (2020), Meulemans et al. (2020, 2021)], the relationship between the proposed method and the Gauss-Newton method seems obvious.
- The advantages over existing target propagation methods are the selling point of this paper, but the experiments in the main text are conducted for comparing only with back-propagation. Moreover, the empirical result is not so convincing because the performance is not evaluated for the NLP tasks, which are the standard application of RNNs.


**Minor comments**
- On page 6: the inequality $\mathbb{E}_{z\sim \mathcal{N}(0,\sigma^2I)}[a^{-1}...] \leq \ell_a ...$ seems wrong. It would probably be a typo.
- The summation regarding the time-step seems missing in the first result in Lemma 3.1.

---

> ### Author Response · Authors · 2022-09-03
> **Answer to Reviewer 9aYY**
>
> Thank you for your detailed and insightful comments! Below we detail the changes we have done in our manuscript (highlighted in teal) following your suggestions, and we answer your comments.
> - Concerning the evaluation in terms of computational efficiency, we presented in Fig. 13 in Appendix D.2 the comparison of TP against BP in terms of overall time, which shows that TP may be advantageous in this case. Note that the goal of our manuscript is to investigate target propagation, an algorithm that inspired several works for some decades, from an optimization viewpoint. We do not claim that target propagation is necessarily better than gradient back-propagation. In Fig. 7, we rather investigate regimes where TP can be advantageous over BP and show that TP does not necessarily outperform BP. Our contribution is to integrate the idea of target propagation in a differentiable framework to evaluate each of its heuristics from an optimization viewpoint on several experiments.
> - Getting convergence rates for TP is a very interesting avenue for future work. We focused on evaluating a meaningful implementation of TP to properly build some theoretical understanding of this method. Our result in Lemma C.5 provides an admittedly simple result. Yet it stands in contrast with previous work, such as the one of Meulemans et al. 2020 that mostly considers asymptotic regimes to draw theoretical interpretations of TP. In particular, the bound we present in Lemma 3.1 and the discussion presented in Appendix C.3 detail every bound in a non-asymptotic regime to build upon.
> - Concerning the relationship with Gauss-Newton, we do not claim that this interpretation is new. On the contrary, we discuss this interpretation in Appendix C.3 and explore it numerically thanks to our implementation of TP using analytical inverses.
> - We added additional comparisons with a previous implementation of TP by Manchev & Spartling, 2020 in Fig. 11 in the Appendix, which shows the potential improvements obtained by considering regularized inversions.
> In Fig. 4 last panel, we added a comparison of TP against BP on the task of predicting words on the Penn Treebank dataset, which shows that our implementation can easily be adapted to this setting while optimizing the training loss correctly in this task.
>
> We thank you again for your comments and the minor details you pointed out that we corrected. Both helped us a lot to give a better perspective of TP and improve our manuscript.

---

> > ### Comment · Action_Editors · 2022-10-27
> > **Responses to minor comments and aim of Lemma 3.1**
> >
> > Dear authors,
> > Following up on your response to reviewer 9aYY, could you please still address the Minor comments as well as the comment regarding the aim of Lemma 3.1?
> > Kind regards,

---

> > > ### Author Response · Authors · 2022-10-29
> > > **Answer to minor comments**
> > >
> > > Dear action editor,
> > >
> > > The minor comments that reviewer 9aYY pointed out were typos that have been corrected in the revised version.
> > > - On page 6 the statement in the original manuscript was written with a wrong inequality, namely, it used <= while it should have read >= which justifies the claim that the regularized inverse minimizes an upper bound of the problem of inverting perturbed layers. It has been corrected.
> > > - On Lemma 3.1, the statement on the original manuscript was using a constant c for all t, while it should have used a constant c_t for each term and a sum over all those terms. This has been corrected too. The final interpretation is essentially the same as our goal was to analyze the differences in the oracles in terms of the differences in the backpropagation scheme at each step to understand the impact of the orthogonality of the weights and the invertibility of the activation function.
> > >
> > > We thank again reviewer 9aYY to point out these minor errors and we are at your disposal for any further discussion.

---

> > > > ### Comment · Reviewer_9aYY · 2022-11-20
> > > > **Thanks for clarification**
> > > >
> > > > Dear authors,
> > > >
> > > > Thanks for your clarification. I confirmed typos are well-corrected.
> > > > Although a concern regarding LemmaC.5 still remains, I would like to acknowledge empirical evidence of the proposed method.

---

### Review · Reviewer_Kkxk · 2022-08-21

**Summary Of Contributions:**

The paper proposes a new approach to target propagation (TP) to train RNNs. For this purpose, they employ regularized inversions of network layers. The proposed approach can be implemented using differentiable programming frameworks. In the theoretical analyses, the relationship between TP and back-propagation (BP) was explored. In the experimental analyses, they examine the proposed TP for several   sequence modelling tasks using RNNs.

**Broader Impact Concerns:**

A Broader Impact Statement was not provided. Ethical implications and its broader impact should be discussed.

**Requested Changes:**

First, the claims of the paper should be revised. The paper propose a new approach for TP for particularly RNNs. For a more general claim, TP should be employed for other DNNs as well. Otherwise, claims should be revised.

Next, according to the revised claims, additional ablation study, additional experiments with other tasks where RNNs have been used successfully, and more detailed comparison with related work should be provided.

In the theoretical analyses, how the proposed TP approach helps convergence of models and resolving vanishing/exploding gradients, and its relation with Gauss-Newton methods, should be elucidated.

**Strengths And Weaknesses:**

The strengths of the paper can be summarized as follows:

- The manuscript is well-written except a few minor problems in the statements. A clear review of the TP and BP was also given in the supplementary material.

- In the theoretical analyses, the stability of the proposed TP, and its relation to the BP was explored.

- The proposed method can be implemented easily with the current libraries with autograd.

- In the initial analyses, the TP performs on par with or outperforms the BP.

One of the main weaknesses of the paper is novelty and claims. Indeed, the paper proposes the method for training RNNs. Therefore, either the proposed method should be extended for training other models (e.g. CNNs), or the title and the claims of the paper should be revised accordingly.

Another major issue is the limited experimental analyses. In the initial results, the proposed method outperforms BP or perform on par in some tasks. However, the benefits of TP are not clear in these analyses. First, ablation study exploring the theoretical results should be given. Then, TP should be employed on other tasks where RNNs have been used as vanilla models, such as speech or text recognition or analysis.

Finally, in the theoretical analyses, the relationship between TP and BP has been analyzed. However, how this helps to resolve other claims such as solving vanishing/exploding gradients and improvement of learning rates, are not addressed. These claims should be further theoretically/experimentally analyzed or revised.

---

> ### Author Response · Authors · 2022-09-03
> **Answer to Reviewer Kkxk**
>
> Thank you for your detailed and insightful comments! Below we detail the changes we have done in our manuscript (highlighted in teal) following your suggestions, and we answer your comments.
> - As suggested, we changed the title and a few sentences in the introduction to reflect that our work investigated TP for RNNs in particular.
> - We added on Fig. 4 last panel the performance of our implementation of TP against gradient back-propagation on a task consisting in predicting words from the Penn Treebank dataset. We presented in Appendix B how we adapted our implementation of TP in this case. We observe that our implementation of TP is still able to decrease the overall training loss while performing reasonably in terms of accuracy.
> - We presented in Fig. 5 the comparison of our implementation of TP with an implementation of TP using parameterized inverses and the difference target propagation formula and an implementation of TP using regularized inverses and the difference target propagation formula. This plot, previously given in the Appendix, can be seen as an ablation study that evaluates the impact of using regularized inverses as opposed to parameterized inverses and linearized propagation as opposed to finite-difference propagation. We observe that the main ingredient that leads to better performance of our implementation of TP is the use of regularized inverses. On the other hand, the linearized formula allows us to easily adapt our implementation to different architectures.
> - We present in Fig. 12 in Appendix D.2 the norm of the displacements along the sequence analyzed by an RNN and compare it to its counterpart when using BP. Interestingly, TP does not necessarily lead to a better propagation of the displacements that can be seen as a counterpart of the gradients computed by BP. This additional experiment gives more insights on TP and helps understand its advantages/disadvantages in contrast with the pure performance plot given in Fig. 4. This additional experiment you and reviewer wpQC help to give a better picture of TP to disentangle beliefs built on previous heuristics used in TP.
> - We have added in Fig. 11 additional comparisons with the previous work of Manchev & Spartling 2020.
> - Concerning the relation with the Gauss-Newton method, we have provided a thorough discussion in Appendix C.3 that details common and divergent points between the methods. In brief, our implementation of TP, using analytical inverses, is able to test the interpretation of TP as a Gauss-Newton method by ignoring the regularization term. However, such an approach did not work experimentally, as explained with the help of Fig. 6, which suggests that the interpretation of TP as a Gauss-Newton method may not lead to a successful implementation.
> - This work investigates an optimization algorithm for deep learning; its ethical implications are not immediate. We are happy to add such a statement in the paper if the reviewer considers it necessary. We believe our conclusion points to future directions and, therefore, broader impact.
>
> We thank you again for your insightful review that helped us give a better overview of TP and improve our manuscript.

---

### Author Response · Authors · 2023-01-09
**Minor Revision and Camera Ready Version**

Dear Action Editor and Reviewers,

Thank you very much for your time and consideration in reviewing this paper! We are thankful for the modifications you proposed that helped improve the paper. We have uploaded a camera-ready version with the following minor revisions.

We added experiments in the Appendix (Figure 13) to further investigate the behavior of the norm of the oracle directions between TP and BP. We also added a reference to a recent paper tackling a similar approach, namely, "LocoProp: Enhancing BackProp via Local Loss Optimization" by Amid et al. Finally, we reformatted the paper for the camera-ready version and added acknowledgments.

We are at your disposal for any further action.

Thank you again for all your work.

Vincent, Zaid

---

### Decision · Action_Editors · 2022-11-20

**Recommendation:** Accept with minor revision

**Comment:**

The article is very well written and in particular it provides a detailed introduction to the subject of target propagation. The reviewers had some concerns about the motivation and correctness of some of the statements, several of which could be clarified during the discussion. Following the rebuttal and discussion, reviewers found that the submission improved in important ways. Particularly, they acknowledged the empirical evidence in support of the proposed method, although it was also mentioned that further convergence analysis (with concerns about the significance and novelty of Lemma C.5 as well as Corollary 4), as well as experiments on different tasks would have made this a stronger contribution.

Overall, reviewers lean accept. I find that the article addresses interesting topics and provides adequate evidence in support of their main claims. Hence I am recommending accept. In consideration of the above remarks, I still encourage a minor revision.




**Audience:**

The topic and findings are of interest to at least some of the individuals in TMLR's audience. The article is also very well written.

**Claims And Evidence:**

The initial submission led to discussions. The authors clarified some inconsistencies that had been pointed out by the reviewers. The reviewers found that the article improved in the revision. Although additional work on convergence results and experiments on different tasks could further strengthen the work, it provides sufficient evidence in support of its claims.